# PRPO: Paragraph-level Policy Optimization for Vision-Language Deepfake Detection

Tuan Nguyen [1]   Naseem Khan [1]   Khang Tran [2]   Hai Phan [2]   Issa Khalil [1]

## Abstract

The rapid rise of synthetic media has made deepfake detection a critical challenge for online safety and trust. Progress remains constrained by the scarcity of large, high-quality datasets. Although multimodal large language models (LLMs) exhibit strong reasoning capabilities, their performance on deepfake detection is poor, often producing explanations that are misaligned with visual evidence or hallucinatory. To address this limitation, we introduce a reasoning-annotated dataset for deepfake detection and propose Paragraph-level Relative Policy Optimization (PRPO), a reinforcement learning algorithm that aligns LLM reasoning with image content at the paragraph level. Experiments show that PRPO improves detection accuracy by a wide margin and achieves the highest reasoning score of 4.55/5.0. Ablation studies further demonstrate that PRPO significantly outperforms GRPO under test-time conditions. These results underscore the importance of grounding multimodal reasoning in visual evidence to enable more reliable and interpretable deepfake detection.

## 1. Introduction

Generative Artificial Intelligence (GAI) has advanced rapidly with the development of generative adversarial networks (GANs) (Goodfellow et al., 2014), diffusion models (Ho et al., 2020; Song et al., 2021a), and their variants (Song et al., 2021b; Rombach et al., 2022; Ho & Salimans, 2021; Ho et al., 2022; Salimans & Ho, 2022). These models, based on distribution matching, generate high-quality synthetic samples that support a wide range of applications (Isola et al., 2017; Karras et al., 2019; Patash-

nik et al., 2021; Rombach et al., 2022; Ho et al., 2022). However, the same capability has fueled deepfake creation, where real and synthetic images are nearly indistinguishable (Nightingale & Farid, 2022; Lu et al., 2023), enabling misuse in misinformation, identity theft, and challenges to authorship (Korshunov & Marcel, 2018; Mirsky & Lee, 2021; Kietzmann et al., 2020; Nguyen et al., 2022). Detecting deepfakes is particularly challenging because synthetic data is trained to mimic the real distribution, making decision boundaries subtle and unstable, especially for unseen models or domains. This motivates the development of detection methods that move beyond surface artifacts to leverage deeper, semantically meaningful cues for robust real-fake discrimination.

Deepfake technology has advanced rapidly, raising the bar for detection methods to be practically useful. Beyond binary classification, reliable reasoning for why an image is flagged as a deepfake is increasingly critical (Zhang et al., 2024). Multimodal Large Language Models (MLLMs) show promise for this task (Liu et al., 2023; Touvron et al., 2023; OpenAI, 2023; Team et al., 2025a), as their large-scale training enables them to capture semantic structures and global visual meaning. However, several challenges limit their effectiveness for deepfake detection. (i) **Data requirements:** detection is data-hungry, and fine-tuning via simple question-answer distillation (Chen et al., 2026) is inadequate, since models such as LLaVA (Liu et al., 2023) are optimized for short answers rather than detailed reasoning. (ii) **Architecture limitations:** subtle manipulations demand fine-grained visual representations that generic vision encoders often fail to capture. (iii) **Reasoning quality:** existing models frequently jump to premature conclusions, biasing subsequent reasoning and overlooking visual evidence. Ensuring systematic inspection of artifacts and progressive reasoning before prediction is thus essential for reliable detection. Figure 1 illustrates these limitations by contrasting explanations from LLaVA with those of our proposed PRPO.

In this paper, we aim to improve both the generalization ability of deepfake detection and the quality of its reasoning responses. Our main contributions are as follows:

---

[1]Qatar Computing Research Institute, Hamad Bin Khalifa University, Doha, Qatar [2]New Jersey Institute of Technology, NJ, USA. Correspondence to: Tuan Nguyen <ntuan@hbku.edu.qa>.

*Proceedings of the $43^{rd}$ International Conference on Machine Learning*, Seoul, South Korea. PMLR 306, 2026. Copyright 2026 by the author(s).

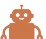
**LLaVA**

The image appears to be a real photograph of a woman with blonde hair. There are no signs of digital manipulation or artificial generation. The woman is looking directly into the camera, and her facial features, such as her eyes and lips, are clearly visible. The image is not fake or manipulated.

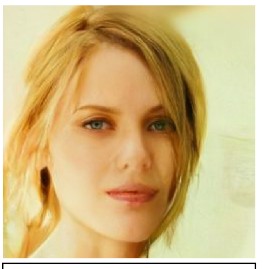

Ground Truth: Fake

**PRPO**

**Skin Texture Anomalies:** The skin in the image exhibits a lack of natural detail, appearing unnaturally smooth and uniform across different areas …
**Background and Surrounding Anomalies:** The background near the face or head shows distortions, anomalies, or inconsistencies that are not consistent with a natural scene …
**Resolution and Sharpness Discrepancies:** The face appears to have a different level of resolution, pixelation, or sharpness compared to the rest of the image …
**General Image Artifacts:** The image contains overall visual noise or artifacts that are not tied to specific features but contribute to a general sense of artificiality or manipulation …
**### Answer:** fake.

*Figure 1.* Reasoning quality comparison between LLaVA and the proposed PRPO. LLaVA and other MLLMs often produce surface-level predictions, yielding misleading reasoning (red) or irrelevant descriptions unrelated to deepfake detection (blue). In contrast, PRPO generates visually grounded explanations (green), describing each deepfake characteristic in a dedicated paragraph and systematically aligning reasoning with image evidence before reaching a conclusion.

- We introduce **DF-R5**, a reasoning-annotated dataset for deepfake detection containing 115k images paired with high-quality explanations, designed to enhance the reasoning capabilities of MLLMs. This dataset fills the gap in reasoning annotations and supports community research in this area.

- We design **DX-LLaVA**, a multimodal architecture for Deepfake detection and eXplainability, which integrates a CLIP ConvNeXT (Liu et al., 2022) vision encoder with a Vicuna language model to capture fine-grained visual artifacts while leveraging strong reasoning ability.

- We propose Paragraph-level Relative Policy Optimization (**PRPO**), a novel test-time reinforcement learning algorithm that aligns MLLM reasoning with image content at the paragraph level. PRPO encourages the model to generate explanations that are not only accurate but also detailed to the visual evidence present in the images. Especially, PRPO can be applied at test time, making it a flexible solution for enhancing reasoning without requiring extensive retraining with annotated data.

- We conduct extensive experiments showing significant gains in both detection accuracy and explanation faithfulness. On unseen domains, our method achieves an average improvement of 14.65% and a reasoning score of 4.55/5.0, surpassing Gemini's 4.2/5.0. These results highlight the importance of grounding multimodal reasoning in visual evidence for reliable and interpretable deepfake detection. Code and dataset are available at https://github.com/tuanrpt/PRPO.

## 2. Related Work

**Traditional Deepfake Detection** The rapid progress of generative AI has made distinguishing real from synthetic images a central problem in image forensics. Modern deepfake detection targets images produced by diffusion models (Ho et al., 2020; Song et al., 2021a), GANs (Goodfellow et al., 2014), and related techniques, whose outputs often exhibit photo-realistic quality. Recent approaches transfer powerful vision backbones such as CLIP-ViT (Dosovitskiy et al., 2021) and ConvNeXT (Liu et al., 2022) to detection tasks (Sha et al., 2023; Abdullah et al., 2024; Ojha et al., 2023; Khan et al., 2025; Nguyen et al., 2025), or exploit frequency-domain features to capture subtle generative artifacts (Frank et al., 2020; Jiang et al., 2021; Koo et al., 2024; LI et al., 2024; Ricker et al., 2024; Tan et al., 2024). While effective, these methods largely lack explainability, providing limited insight into the cues driving predictions.

**Deepfake Detection with LLMs** Large Language Models (LLMs) excel in multimodal tasks such as captioning (Radford et al., 2021; Li et al., 2022; 2023), Visual Question Answering (VQA) (Liu et al., 2023; 2024), and even image generation (Rombach et al., 2022; Podell et al., 2024). Recent work has explored fine-tuning Multimodal LLMs (MLLMs) for deepfake detection (Chen et al., 2026; Li et al., 2025; Huang et al., 2024; Xu et al., 2025; Huang et al., 2025). However, existing datasets rarely include detailed reasoning annotations, leading models to generate shallow explanations that overlook critical cues. Moreover, explanation quality is seldom evaluated, limiting trustworthiness. To address this gap, we introduce a reasoning-annotated dataset for deepfake detection and a reinforcement learning framework that trains models to provide accurate, interpretable reasoning.

**Test-Time Reinforcement Learning** Reinforcement Learning (RL) has proven effective for improving LLM outputs (Ouyang et al., 2022; Rafailov et al., 2023; Shao et al., 2024; Zhao et al., 2026). RLHF (Christiano et al., 2017; Ouyang et al., 2022) aligns models with human preferences via algorithms such as PPO (Schulman et al., 2017) and DPO (Rafailov et al., 2023). Group Relative Policy Optimization (GRPO) (Shao et al., 2024) extends this by optimizing relative quality across responses, mitigating reward sensitivity and improving stability, powering models like DeepSeek-R1 (Guo et al., 2025). More recently, Test-Time RL (TTRL) enables models to self-improve during inference using majority voting (Zuo et al., 2026) or self-certainty rewards (Zhao et al., 2026), without additional training data. RL has also been applied in multimodal tasks such as captioning (Ren et al., 2017; Zhang et al., 2025a) and VQA (Zhang et al., 2025b; Xia et al., 2025), but remains underexplored for deepfake detection. A central challenge is designing rewards that capture both detection accuracy and fine-grained explanatory cues, which LLMs often overlook or hallucinate. Our method, PRPO, addresses this by introducing paragraph-level rewards that explicitly align explanations with visual evidence, advancing RL-driven deepfake detection and explainability.

## 3. Methodology

In this section, we introduce the DF-R5 dataset, a large-scale multimodal deepfake reasoning corpus constructed from state-of-the-art multimodal large language models (MLLMs) for deepfake detection. We then refine the quality of its reasoning annotations using the Paragraph-level Relative Policy Optimization (PRPO) algorithm.

### 3.1. Reasoning Data Generation

*Table 1.* MLLM performance (%) on 1,000 DF-R5 samples.

| Model | Acc | Pre | Rec | F1 |
|---|---|---|---|---|
| Claude-3 | 50.80 | 65.38 | 3.40 | 6.46 |
| Pixtral | 51.60 | 71.05 | 5.40 | 10.04 |
| LLaMA-4 | 64.90 | 73.21 | 47.00 | 57.25 |
| Qwen-2.5 | 62.64 | 69.17 | 52.23 | 59.52 |
| GPT-4o | 70.80 | **93.33** | 44.80 | 60.54 |
| Gemini-2.5 | **77.60** | 75.09 | **82.60** | **78.67** |

DF-R5 is a multi-domain deepfake reasoning dataset containing approximately 115k image-reasoning pairs with rich semantic annotations. The base images are sourced from DF40 (Yan et al., 2024), covering five diverse generative domains: DDIM (Song et al., 2021a), PixArt-$\alpha$ (Chen et al., 2024), SD-2.1 (Rombach et al., 2022), SiT (Ahmed et al., 2021), and StyleGAN3 (Karras et al., 2021). These domains are selected to maximize both diversity and difficulty, ensuring robust generalization across generation methods. A

naive approach would be to directly distill reasoning from MLLMs using the collected images. However, two challenges arise: (1) identifying which MLLM provides the strongest reasoning quality, and (2) designing prompting strategies that can minimize hallucination and misinformation.

To address the first challenge, we systematically benchmark several representative MLLMs by asking them to classify 1,000 randomly selected images (balanced between real and fake). We chose well-known MLLMs including Claude-3 (Anthropic, 2024), Pixtral (Agrawal et al., 2024), LLaMA-4 (Meta AI, 2025), Qwen-2.5 (Qwen et al., 2025), GPT-4o (OpenAI, 2023), Gemini-2.5 (Team et al., 2025a). The results, presented in Table 1, indicate that Claude-3 and Pixtral yield high precision but fail to capture most true cues, resulting in very low recall. Gemini-2.5 achieves the best trade-off, with the highest overall accuracy (77.60%) and F1 score (78.67%), and is thus selected as the primary model for reasoning distillation.

To address the second challenge, we design a three-step pipeline, illustrated in Figure 2, to generate consistent, high-quality reasoning annotations from Gemini:

**Step 1: Feature Discovery.** We prompt multiple vision-language models (Gemini-2.5, Qwen-2.5, LLaMA-4 and GPT-4o) to enumerate facial and visual characteristics relevant to deepfake detection. Each model proposes $K$ candidate features (e.g., $K = 50$), yielding approximately $4 \times K$ (e.g., 200) unique features in total. Importantly, no images are provided at this stage; the prompts focus on eliciting general, commonly recognized features from the models. After deduplication and consolidation, we curate a final set of $k$ features (e.g., $k = 74$).

**Step 2: Feature Scoring.** For each image, given the list of $k$ features, we prompt Gemini to assign a score of `Real` ($-1$), `Fake` ($+1$), or `Uncertain` ($0$) to each feature. This step mitigates the risk of Gemini selecting all features indiscriminately or relying on hallucinated ones among the $k$ candidates. Cases flagged as incorrect or uncertain are further refined through additional prompting with ground-truth labels. This procedure enhances the reliability of the predictions, as suggested by (Zelikman et al., 2022). The distribution of the collected features is reported in Appendix A.1.

**Step 3: Reasoning Generation.** Each image now has a corresponding set of feature scores from Step 2. To avoid redundancy and overly long explanations, we instruct Gemini to consolidate fine-grained feature scores into at most $M$ semantically coherent groups (e.g., $M = 7$). The choice of $M$ is guided by the 85% group-frequency threshold, ensuring that the majority of commonly observed features are represented. Importantly, we do not require Gemini to map features into a fixed set of groups; instead, it is prompted to

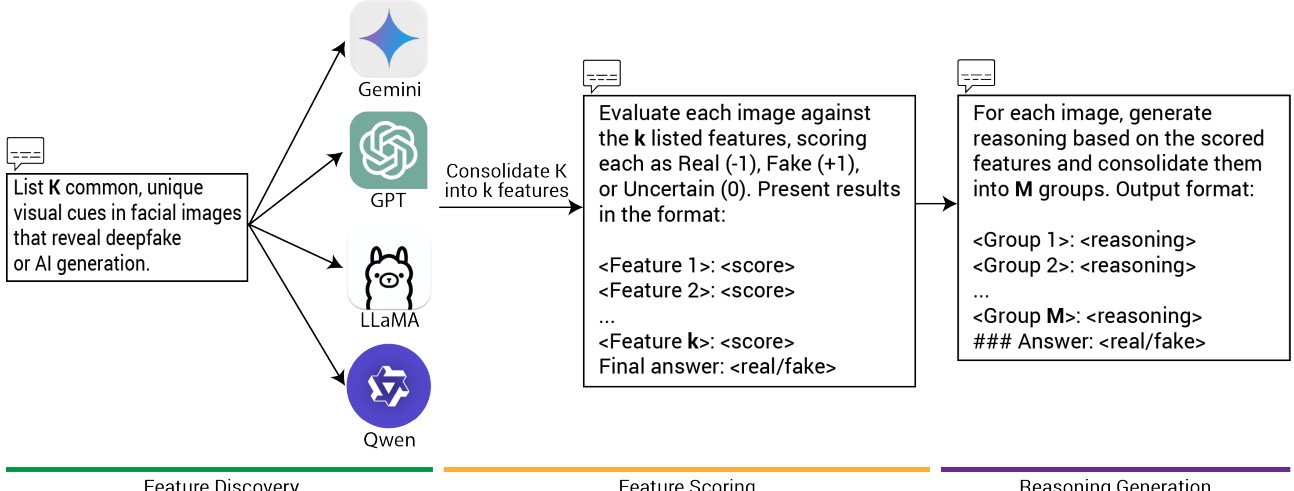

*Figure 2.* Three-step pipeline for generating high-quality reasoning annotations in DF-R5. The detailed prompts for each step are presented in the Appendix A.2.

organize them into at most $M$ groups depending on the content of each image. This produces concise and interpretable reasoning descriptions for each image. The full prompt templates for each step are provided in Appendix A.2.

### 3.2. Fine-tuning with DX-LLaVA

*Table 2.* Intra-domain vs. inter-domain performance (%) for fine-tuned LLaVA.

| Method | Acc | Prec | Rec | F1 |
|---|---|---|---|---|
| Intra-domain | 99.57 | 99.84 | 99.35 | 99.59 |
| Inter-domain | 59.40 | 98.26 | 21.90 | 35.82 |

In this section, we fine-tune a LLaVA-based architecture (Liu et al., 2023) on our DF-R5 dataset. We start with naive fine-tuning and progressively incorporate enhancements to improve generalization across unseen domains.

Our baseline is the original LLaVA, comprising a CLIP ViT-L/14 visual encoder (Radford et al., 2021), a Vicuna language model (Chiang et al., 2023), and a multimodal projector mapping ViT patch embeddings into Vicuna's token space. Table 2 reports results under two settings: (i) **Intra-domain**, with random train/validation/test splits, and (ii) **Inter-domain**, with one domain held out. Details about experimental setup are described in Appendix E.3. While intra-domain accuracy exceeds 99%, inter-domain performance drops sharply: precision remains high (98.26%), but the model collapses to predicting nearly all images as real. Moreover, we find that Vicuna generates coherent text yet fails to distinguish real from fake, reflecting poor alignment of the projector with discriminative visual cues. To address this, we add a lightweight classifier on top of the projector.

CLIP patch embeddings are aggregated via global average pooling (GAP) into a pooled representation $\bar{e}$, which is then classified:

$$e = \text{CLIP}(x) \in \mathbb{R}^{P \times d}, \bar{e} = \text{GAP}(e), \hat{y} = \mathcal{C}(\bar{e}; \phi), \quad (1)$$

where $e$ denotes the patch embeddings, and $\mathcal{C}$ the classifier parameterized by $\phi$. We now train the classifier along with the projector $W$ and Vicuna $\pi_\theta$ by minimizing the following total loss $\mathcal{L}_{\text{total}}$:

$$\mathcal{L}_{\text{lm}} = \mathbb{E}_{(x, o_{<t}, o_t) \sim \mathcal{D}} \big[ - \log \pi_\theta(o_t \mid o_{<t}, z) \big], \quad (2)$$

$$\mathcal{L}_{\text{binary}} = \mathbb{E}_{(x,y) \sim \mathcal{D}} \big[ - y \log \hat{y} \big], \quad (3)$$

$$\min_{\theta, W, \phi} \mathcal{L}_{\text{total}} = \mathcal{L}_{\text{lm}} + \alpha \mathcal{L}_{\text{binary}}, \quad (4)$$

where $\mathcal{D} = \{(x_i, o_i, y_i)\}_{i=1}^N$ is our dataset consisting of images $x_i$, reasoning sequences $o_i$, and corresponding ground-truth labels $y_i$. Here, $z = e \cdot W$ denotes the projected image token input to Vicuna, and $\alpha$ is a trade-off parameter that balances the language modeling loss $\mathcal{L}_{\text{lm}}$ with the binary classification loss $\mathcal{L}_{\text{binary}}$.

*Table 3.* Performance comparison (%) of $\mathcal{L}_{\text{lm}}$ and $\mathcal{L}_{\text{lm}} + \alpha \mathcal{L}_{\text{binary}}$.

| Method | Acc | Pre | Rec | F1 |
|---|---|---|---|---|
| $\mathcal{L}_{\text{lm}}$ | 59.40 | 98.26 | 21.90 | 35.82 |
| $\mathcal{L}_{\text{lm}} + \alpha \mathcal{L}_{\text{binary}}$ | 70.40 | 92.97 | 46.12 | 61.66 |

Table 3 shows that incorporating a binary loss improves detection performance and strengthens Vicuna's ability to discriminate between real and fake images. However, gains remain limited by a core weakness of LLaVA: the CLIP ViT encoder (Radford et al., 2021). While ViT captures global

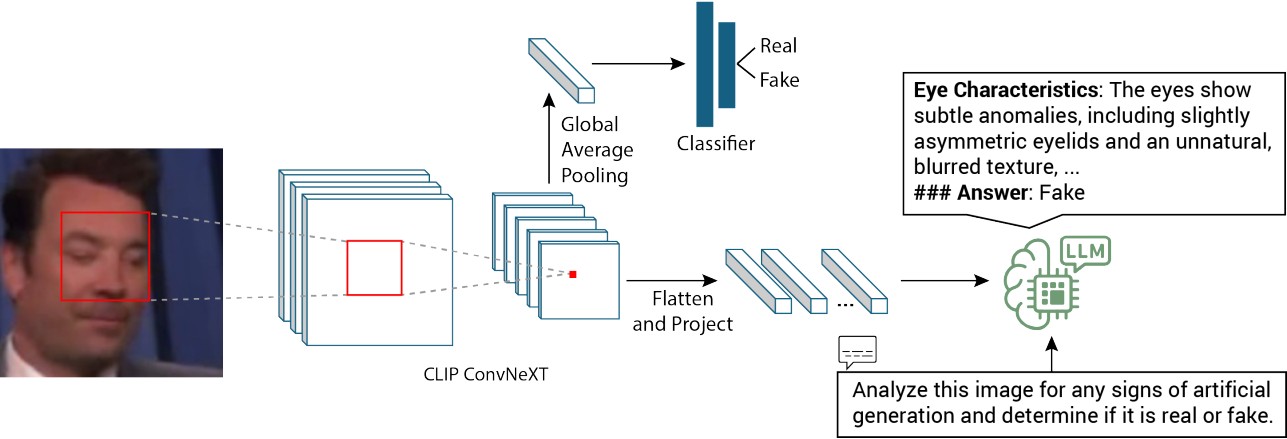

*Figure 3.* Proposed DX-LLaVA, a LLaVA fine-tuning framework for **D**eepfake detection and e**X**plainability. Unlike CLIP ViT, which outputs patch embeddings, CLIP ConvNeXT produces pixel-level embeddings. This enables a finer focus on local image regions, leading to improved deepfake detection and reasoning performance.

semantics effective for VQA, deepfake detection demands sensitivity to local, high-frequency artifacts, making ViT suboptimal. To overcome this, we propose **DX-LLaVA**, which replaces CLIP ViT with CLIP ConvNeXT (Liu et al., 2022), a convolutional encoder with stronger texture bias and greater sensitivity to subtle artifacts such as hairline irregularities, pore inconsistencies, and abnormal background details (Figure 3). Specifically, we use the output from **Stage 3** of ConvNeXT, yielding a $10 \times 10$ feature map flattened into 100 pixel embeddings and projected into Vicuna's text embedding space via the projector. The objective remains as in Eq. (4), with both the projector and Vicuna fine-tuned to adapt to ConvNeXT features. The effectiveness of DX-LLaVA is examined in Section 4.7. Detailed comparison between CLIP ViT and CLIP ConvNeXT is also provided in Appendix C.2.

### 3.3. Test-time Deepfake Detection with PRPO

After fine-tuning, we observed two recurring issues in the generated reasoning: (i) **Image-Reason Consistency**, where explanations often failed to align with visual content, including redundant or overly generic cues not present in the image; and (ii) **Reason-Prediction Consistency**, where the final answer occasionally contradicted the consensus of supporting paragraphs, producing incorrect outputs despite consistent evidence. These issues stem from the fact that MLLMs are generally optimized for the final decision, which can induce hallucinations in intermediate reasoning. Existing RL algorithms also focus on rewarding final outputs (Shao et al., 2024), often neglecting reasoning quality. Unlike mathematical reasoning, where step-wise evaluation is feasible (Wang et al., 2024; Cui et al., 2025), evaluating intermediate textual reasoning in deepfake detection is more challenging. To address this gap, we propose a reinforcement learning algorithm that mitigates misinformation, en-

forces consistency between reasoning and predictions, and yields more reliable explanations for deepfake detection.

We propose a novel Paragraph-level Relative Policy Optimization (PRPO), inspired by the GRPO algorithm (Shao et al., 2024). Given a prompt $v$ and image tokens $z$, the policy $\pi_\theta$ produces a set of sampled outputs $\mathcal{O} = \{o^{(1)}, o^{(2)}, \ldots, o^{(L)}\}$. Each output $o^{(i)}$ is split into $M_i + 1$ paragraphs: $o^{(i)} = \{p_1^{(i)}, p_2^{(i)}, \ldots, p_{M_i+1}^{(i)}\}$, where $p_{M_i+1}^{(i)}$ denotes the final-answer sentence. To enhance the reasoning ability of DX-LLaVA, we introduce two reward functions: the Visual Consistency Reward (VCR) and the Prediction Consistency Reward (PCR).

**Visual Consistency Reward (VCR).** VCR enforces alignment between each paragraph $p_j^{(i)}$ and the image features. Specifically, we leverage the frozen CLIP ConvNeXT encoder in our architecture to compute alignment between image features and paragraph content. Since CLIP ConvNeXT is limited by input length, we first extract representative keywords from each paragraph using the YAKE library[1] (Campos et al., 2020): $s_j^{(i)} = \text{YAKE}(p_j^{(i)})$. YAKE is a lightweight, unsupervised keyword extraction method requiring no external models. The reward score is then computed as:

$$R_{\text{VCR}}(p_j^{(i)}) = \tfrac{1}{2}\big[\text{sim}\big(\text{CLIP}_{\text{txt}}(s_j^{(i)}), \text{CLIP}_{\text{img}}(x)\big) + 1\big],$$
(5)

where $\text{sim}(\cdot, \cdot)$ denotes cosine similarity, $\text{CLIP}_{\text{txt}}(.)$ and $\text{CLIP}_{\text{img}}(.)$ are the text and image embeddings produced by the frozen CLIP ConvNeXT encoder, and $R_{\text{VCR}}$ is normalized to $[0, 1]$. The effectiveness of VCR is further investigated in Appendix B.1.

**Prediction Consistency Reward (PCR).** PCR evaluates the

---

[1]https://github.com/LIAAD/yake

internal consistency between the majority vote of reasoning paragraphs and the final conclusion sentence $p_{M_i+1}^{(i)}$. In our dataset, we observe that all reasoning paragraphs typically agree when describing real or fake characteristics. Therefore, for intermediate paragraphs we set $R_{PCR}\left(p_j^{(i)}\right) = 1$ for simplicity. For the final paragraph, however, its reward is determined by whether its prediction matches the majority vote from the preceding paragraphs. We define the majority vote as:

$$\hat{y}_{\mathrm{maj}}^{(i)} = \arg\max_{\ell \in \{real, fake\}} \left| \{ j < M_i + 1 : \hat{y}(p_j^{(i)}) = \ell \} \right|. \quad (6)$$

Here, $\hat{y}(p_j^{(i)})$ denotes the label predicted for paragraph $p_j^{(i)}$ using predefined dictionaries of deepfake-related terms $\mathcal{F} = \{unnatural, inconsistent, manipulated, overly smooth, \dots\}$, real terms $\mathcal{R} = \{authentic, genuine, realistic, natural, \dots\}$, and negation terms $\mathcal{N} = \{no, not, without, lack of, \dots\}$. Then the reward for the final paragraph is:

$$R_{PCR}\left(p_{M_i+1}^{(i)}\right) = \begin{cases} 1.0 & \text{if } \hat{y}(p_{M_i+1}^{(i)}) = \hat{y}_{\mathrm{maj}}^{(i)}, \\ 0.0 & \text{otherwise.} \end{cases} \quad (7)$$

The procedure for reward computation is presented in Algorithms 1, 2, and 3 in Appendix B.2. We find that this approach effectively mitigates the consistency problem while avoiding reliance on external models and additional computational overhead. The overall reward for a paragraph is defined as the average of the two reward components:

$$R\left(p_j^{(i)}\right) = \tfrac{1}{2}\left(R_{VCR}\left(p_j^{(i)}\right) + R_{PCR}\left(p_j^{(i)}\right)\right). \quad (8)$$

where $j = 1, 2, ..., M_i + 1$. The core idea of our reward function design is to leverage **self-consistency** and **visual grounding** without requiring paragraph-level supervision. Importantly, the final prediction does not depend on ground-truth labels. By employing such label-free rewards, the model is encouraged to generalize more effectively to unseen images. For each group $\mathcal{O}$, we compute the mean $\mu_R$ and standard deviation $\sigma_R$ of rewards across all paragraphs. The normalized relative advantage of paragraph $p_j^{(i)}$ is defined as $A_j^{(i)} = \frac{R\left(p_j^{(i)}\right) - \mu_R}{\sigma_R + \epsilon}$, where $\epsilon$ is a small constant added for numerical stability.

Given a prompt $v$ and image tokens $z$, PRPO maximizes the log-probabilities of paragraphs weighted by their own relative advantage, with PPO-style clipping for stability. We define the policy ratio:

$$r_j^{(i)} = \frac{\pi_\theta\left(p_j^{(i)} \mid v, z\right)}{\pi_{\mathrm{old}}\left(p_j^{(i)} \mid v, z\right)}. \quad (9)$$

The PRPO loss is then:

$$\mathcal{L}_{\mathrm{PRPO}}(\theta) = \mathbb{E}_{\mathcal{O} \sim \pi_\theta} \left[ \sum_{i=1}^{L} \sum_{j=1}^{M_i+1} \right.$$
$$\left. \min\left( r_j^{(i)} A_j^{(i)}, \ \mathrm{clip}(r_j^{(i)}, 1 - \epsilon, 1 + \epsilon) A_j^{(i)} \right) \right]. \quad (10)$$

Different from GRPO, where each token is treated with the same advantage, PRPO computes advantages at the paragraph level. This increases the likelihood of paragraphs (deepfake characteristics) that align with the image and remain consistent with the final answer, while decreasing the likelihood of those that are misaligned. The optimization further incorporates a Kullback-Leibler (KL) divergence loss to encourage exploration of novel reasoning traces at the paragraph level while constraining the policy from deviating excessively from the reference model.

$$\mathcal{L}_{\mathrm{KL}}(\theta) = \frac{1}{\sum_{i=1}^{L}(M_i + 1)} \sum_{i=1}^{L} \sum_{j=1}^{M_i+1}$$
$$\mathbb{E}_{p_j^{(i)} \sim \pi_\theta} \left[ \log \frac{\pi_\theta(p_j^{(i)} | v, z)}{\pi_{\mathrm{ref}}(p_j^{(i)} | v, z)} \right]. \quad (11)$$

The overall optimization objective combines the PRPO loss with the KL regularization term:

$$\max_\theta \ \mathcal{J}_{\mathrm{total}}(\theta) := \mathcal{L}_{\mathrm{PRPO}}(\theta) - \beta \, \mathcal{L}_{\mathrm{KL}}(\theta), \quad (12)$$

where $\beta$ is a weighting coefficient.

## 4. Experiment

### 4.1. Dataset

We construct our dataset from DF-40 (Yan et al., 2024), which integrates widely used deepfake benchmarks such as FaceForensics++ (Rossler et al., 2019), and CelebDF (Li et al., 2020). To generate synthetic images, we employ generative models including DDIM (Song et al., 2021a), PixArt (Chen et al., 2024), SD-2.1 (Rombach et al., 2022), SiT-XL/2 (Ahmed et al., 2021), and StyleGAN (Karras et al., 2021). For each domain, we collect 30k images, resulting in 150k samples in total. After filtering invalid formats with Gemini, the final dataset contains around 115k images.

### 4.2. Implementation Details.

We employ full fine-tuning for DX-LLaVA by optimizing the objective in Eq. (4) with $\alpha = 10.0$, and apply PRPO with $\beta = 0.01$ in Eq. (12). A pretrained LLaVA-7B is used with a frozen CLIP ConvNeXT backbone. Images are resized to $320 \times 320$ and processed through the default CLIP pipeline (Radford et al., 2021). The 1536-d CLIP

*Table 4.* Detection performance (F1 score, %) of our method versus baselines. $\rightarrow X$ denotes testing on unseen domain $X$, with the remaining four domains used for training.

| Method | $\rightarrow$ DDIM | $\rightarrow$ PixArt | $\rightarrow$ SD | $\rightarrow$ SiT | $\rightarrow$ StyleGAN | Average |
|---|---|---|---|---|---|---|
| LLaVA | 49.86 | 65.46 | 26.54 | 15.36 | 57.03 | 42.85 |
| DE-FAKE | 8.83 | 86.45 | **95.80** | 4.55 | 76.50 | 54.43 |
| FakeShield | 31.84 | 88.57 | 92.28 | 33.22 | 98.70 | 68.92 |
| UnivCLIP | 74.85 | **89.31** | 74.81 | 40.01 | 86.46 | 73.09 |
| SIDA | 70.07 | 73.86 | 92.37 | 46.53 | 94.98 | 75.26 |
| **DX-LLaVA (ours)** | 92.34 | 83.11 | 89.35 | 26.46 | 99.13 | 78.08 |
| **PRPO (ours)** | **95.88** | 88.10 | 94.99 | **71.26** | **99.32** | **89.91** |

pixel embeddings are projected to 4096-d to align with Vicuna's text space. We set the learning rate to $2 \times 10^{-5}$ for fine-tuning and $3 \times 10^{-7}$ for PRPO reasoning, using the `verl` package (Sheng et al., 2025). All experiments are conducted on 8 NVIDIA H200 GPUs (143 GB each) with AdamW (Loshchilov & Hutter, 2019), enabling parallel reward computation and YAKE-based token extraction. The architecture of DX-LLaVA is described in Appendix C.1.

### 4.3. Generalization Results

Table 4 reports detection accuracy against deepfake detection baselines, including LLaVA (Liu et al., 2023), DE-FAKE (Sha et al., 2023), FakeShield (Xu et al., 2025), UnivCLIP (Ojha et al., 2023), and SIDA (Huang et al., 2025). DX-LLaVA achieves an average F1 of 78.08%, outperforming state-of-the-art baselines (e.g., improving upon SIDA by 2.82%). Incorporating PRPO boosts performance to 89.91%, setting a new state of the art and outperforming SIDA by 14.65%. PRPO is particularly effective in challenging domains such as SiT where real and fake images are nearly indistinguishable, demonstrating its robustness. Additional metrics (accuracy, precision, recall) are provided in Table 17 in the Appendix. Table 5 compares our method against recent MLLMs, including LLaMA-4 Maverick (Meta AI, 2025), Pixtral-12B (Agrawal et al., 2024), Qwen2.5-VL-32B (Qwen et al., 2025), GPT-4o (OpenAI, 2023), Gemma-3-27B-IT (Team et al., 2025b), and Gemini-2.5 (Team et al., 2025a). Although DX-LLaVA benefits from Gemini distillation, its accuracy remains 2% below Gemini-2.5, suggesting that larger-scale data or stronger reasoning may be required. In contrast, PRPO achieves 89.91%, significantly surpassing MLLM baselines, with Gemini-2.5 the closest competitor. These results demonstrate the effectiveness of PRPO in advancing deepfake detection beyond both state-of-the-art baselines and MLLMs.

### 4.4. Explanation Quality Evaluation

Table 6 presents the explanation quality scores assigned by GPT-4o across five key criteria. For each criterion, GPT-4o

*Table 5.* Average detection performance (%) across five domains.

| Model | Acc | Pre | Rec | F1 |
|---|---|---|---|---|
| LLaMA-4 Maverick | 53.58 | 73.29 | 14.59 | 23.23 |
| Pixtral-12B | 63.27 | 69.61 | 36.85 | 44.38 |
| Qwen2.5-VL-32B | 59.54 | 64.87 | 59.54 | 54.18 |
| GPT-4o | 57.96 | 73.38 | 57.96 | 63.11 |
| Gemma-3-27B-IT | 66.44 | 65.69 | 70.55 | 67.64 |
| Gemini-2.5 | 85.00 | 94.23 | 74.44 | 80.31 |
| **DX-LLaVA (ours)** | 84.64 | **99.57** | 70.58 | 78.08 |
| **PRPO (ours)** | **89.02** | 91.40 | **89.42** | **89.91** |

*Table 6.* Reasoning quality evaluation conducted by GPT-4o.

| Model | CAC | EGIA | RQ | CC | CU | Overall |
|---|---|---|---|---|---|---|
| GPT-4o-Mini | 2.98 | 1.32 | 1.94 | 2.67 | 2.99 | 2.38 |
| LLaVA-Base | 3.07 | 2.76 | 2.92 | 3.18 | 3.94 | 3.17 |
| LLaMA-4-Maverick | 2.70 | 3.34 | 3.26 | 3.01 | 4.06 | 3.27 |
| Pixtral-12B | 3.17 | 3.51 | 3.40 | 3.38 | 4.20 | 3.53 |
| Qwen2.5-VL-32B | 3.02 | 3.70 | 3.64 | 3.28 | 4.29 | 3.59 |
| Gemma-3-27B-IT | 3.36 | 3.77 | 3.76 | 3.51 | 4.42 | 3.76 |
| Gemini-2.5 | 3.98 | 4.16 | 4.23 | 4.05 | 4.60 | 4.20 |
| **DX-LLaVA (ours)** | 3.78 | 3.99 | 4.04 | 3.98 | 4.29 | 4.02 |
| **PRPO (ours)** | **4.42** | **4.56** | **4.58** | **4.50** | **4.69** | **4.55** |

rates the explanations on a scale from 1 to 5: (i) **Classification Accuracy and Consistency (CAC)**: correctly classifying the image as real or fake while remaining consistent with the ground truth; (ii) **Evidence Grounding and Image Alignment (EGIA)**: citing visual artifacts that are actually present in the image and avoiding hallucinations; (iii) **Reasoning Quality (RQ)**: providing step-by-step explanations free of contradictions or irrelevant details; (iv) **Confidence Calibration (CC)**: expressing confidence at a level appropriate to the evidence, without overstating or understating certainty; and (v) **Clarity and Usefulness (CU)**: producing clear, well-structured, and interpretable explanations that are useful for human investigators. The detailed prompt is shown in Appendix D.2. We compare our method against several MLLM baselines. Our fine-tuned DX-LLaVA model achieves an overall score of 4.02, out-

*Table 7.* Performance comparison (%) of different reward components on transfer task → **SD**.

| Method | Acc | Prec | Rec | F1 |
|---|---|---|---|---|
| VCR only | 89.20 | 99.30 | 80.11 | 88.68 |
| PCR only | 76.20 | **99.66** | 55.11 | 70.98 |
| Full rewards | **94.80** | 96.67 | **93.37** | **94.99** |

*Table 9.* Ablation study on LLaVA, DX-LLaVA, and PRPO under the inter-domain setting.

| Architecture | PRPO | Acc | Pre | Rec | F1 |
|---|---|---|---|---|---|
| LLaVA | – | 59.40 | 98.26 | 21.90 | 35.82 |
| LLaVA | ✓ | **62.70** | **98.72** | **29.33** | **45.23** |
| DX-LLaVA | – | 92.60 | **99.11** | 86.43 | 92.34 |
| DX-LLaVA | ✓ | **95.80** | 98.79 | **93.14** | **95.88** |

*Table 8.* Performance comparison (%) of different RL methods on transfer task → **DDIM**.

| Method | Acc | Pre | Rec | F1 |
|---|---|---|---|---|
| DX-LLaVA (ours) | 92.60 | 99.11 | 86.43 | 92.34 |
| PPO (TTRL) | 94.00 | 99.79 | 88.76 | 93.95 |
| GRPO (TTRL) | 92.29 | **100.00** | 85.33 | 92.09 |
| **PRPO (ours)** | **95.80** | 98.79 | **93.14** | **95.88** |

performing most baselines except Gemini-2.5 (4.20). With PRPO, the score rises substantially to 4.55/5.0, surpassing all baselines by a notable margin. PRPO shows particular strength in CAC and EGIA, highlighting its ability to align reasoning with image features and maintain consistency in prediction, thereby improving deepfake detection accuracy and producing high-quality, reliable explanations. We also conduct a human-evaluation study which shows similar performance, as detailed in Appendix D.1.

### 4.5. Ablation Study on Reward Components

Table 7 reports ablation results on reward components for transfer to the **SD** domain. Using only the Visual Consistency Reward (VCR) achieves very high precision (99.30%) but low recall (80.11%), while using only the Prediction Consistency Reward (PCR) further reduces recall to 55.11%. These results show that each reward alone is insufficient: VCR mitigates hallucinations but lacks decision-level alignment, whereas PCR enforces alignment but encourages overly generic or systematically incorrect predictions due to the absence of visual grounding. Combining both rewards yields balanced gains, boosting recall to 93.37% while maintaining strong precision (96.67%), and producing the best F1 score (94.99%). In case artifacts are sparse, and PCR's majority voting suppress minority cues, VCR can simultaneously reward visually grounded paragraphs, so rare but valid artifact descriptions receive higher advantage scores and greater influence during policy updates. The combination of VCR and PCR therefore prevents majority bias from dominating, although extremely sparse scenarios remain challenging. Future work could explore weighted voting based on VCR scores to address such special scenarios.

### 4.6. Ablation Study on Other RL Methods

We compare our PRPO framework with other RL algorithms adapted for test-time optimization, including PPO (Schulman et al., 2017) and GRPO (Shao et al., 2024) from Test-Time Reinforcement Learning (TTRL) (Zuo et al., 2026). In TTRL, the authors apply majority voting at the sample level, using the final answer as the prediction. As shown in Table 8, PPO achieves high precision (99.79%) but lower recall (88.76%), reflecting its tendency to produce overly conservative predictions that miss many true positives. GRPO, on the other hand, achieves perfect precision (100.0%) but with slightly lower recall than PPO, suggesting an even stronger bias toward cautious predictions. In contrast, our PRPO method provides the best overall balance, attaining the highest accuracy (95.80%), recall (93.14%), and F1 score (95.88%), while maintaining strong precision (98.79%). PRPO is specifically designed to capture self-consistency between visual cues and reasoning at the paragraph level, highlighting the importance of fine-grained alignment for reliable deepfake detection and explainability.

### 4.7. Effectiveness of DX-LLaVA and PRPO

To further understand the effectiveness of our DX-LLaVA architecture and PRPO method, we conduct an ablation study on the transfer task → DDIM, as shown in Table 9. The results show that the DX-LLaVA architecture significantly outperforms the baseline LLaVA model, achieving a substantial increase of 33.2% in accuracy (from 59.40% to 92.60%) and 56.52% in F1 score (from 35.82% to 92.34%). This improvement highlights the importance of our architectural modifications, including the integration of pixel-level visual features and the dual-objective fine-tuning strategy, which enhance the model's ability to understand and reason about deepfake forensics. We further apply our PRPO method to both LLaVA and DX-LLaVA, which results in consistent performance gains, with 9.41% and 3.54% increases in F1 score on LLaVA and DX-LLaVA, respectively. PRPO effectively improves reasoning alignment with visual evidence and paragraph-level self-consistency, leading to better deepfake detection performance across different architectures.

*Table 10.* Comparison of ViT-L/14 and ConvNeXT backbones with and without auxiliary binary loss.

| Vision Backbone | Aux. Binary Loss | Intra-Domain F1 | Inter-Domain F1 (DDIM) |
|---|---|---|---|
| DX-LLaVA (ViT-L/14) | without binary loss | 96.74 | 61.21 |
| DX-LLaVA (ViT-L/14) | with binary loss | 96.25 | 67.41 |
| DX-LLaVA (ConvNeXT) | without binary loss | **99.91** | **80.86** |
| DX-LLaVA (ConvNeXT) | with binary loss | **99.76** | **91.54** |

*Table 11.* Comparison of supervised and unsupervised PRPO across test domains.

| Test Domain | Method | Acc | Pre | Rec | F1 |
|---|---|---|---|---|---|
| **DDIM** | PRPO-S | 81.60 | 100.00 | 64.95 | 78.75 |
| | PRPO-U | 95.80 | 98.79 | 93.14 | **95.88** |
| **PixArt** | PRPO-S | 87.49 | 99.76 | 76.74 | 86.74 |
| | PRPO-U | 88.60 | 99.29 | 79.17 | **88.10** |

### 4.8. Performances on architecture changes

To compare the architectural impact of CLIP ViT (Radford et al., 2021) versus CLIP ConvNeXT (Liu et al., 2022), we conduct an analysis in which we change the architecture of the vision model in DX-LLaVA from ConvNeXT to ViT-L/14. Note that DX-LLaVA (ViT-L/14) is not the same as the original LLaVA. Here, we use the second-to-last block of ViT-L/14 that outputs spatial feature maps. All other setups are kept the same for a fair comparison. The results are shown in Table 10.

The results show that ConvNeXT outperforms ViT-L/14 by a large margin in both intra-domain (+3.51% F1) and inter-domain (+24.10% F1) settings, confirming that the architectural choice is critical for deepfake detection. Additional ablation studies comparing the two backbones are provided in Appendix C.2.

### 4.9. Supervised PRPO vs Unsupervised PRPO

To testify the performance of PRPO using ground truth in the reward function, we conducted this ablation to compare **S**upervised PRPO (PRPO-S) and **U**nsupervised PRPO (PRPO-U). In supervised PRPO, the majority voting is replaced by the ground truth. The experiments are trained on four domains and tested on a held-out domain (inter-domain). As shown in Table 11, unsupervised PRPO consistently outperforms supervised PRPO in both the DDIM and PixArt test domains, with improvements from 78.75% to 95.88% F1 on DDIM and from 86.74% to 88.10% F1 on PixArt. This is an interesting behavior: PRPO with label-free rewards in VCR and PCR encourages the model based on what it has seen and to remain self-consistent across all paragraph predictions, making the model more invariant across domains. In contrast, supervised PRPO with ground-truth rewards tends to overfit to domain-specific artifacts or annotation biases, leading to performance degradation on unseen generative models. This observation is similar to findings in unsupervised representation learning, where contrastive or self-supervised objectives often yield stronger cross-domain transfer than fully supervised training.

## 5. Limitations

The approach primarily focuses on improving detection for face-centric deepfakes and does not directly address other types of deepfakes involving natural-scene datasets. Our DF-R5 dataset is distilled from Gemini, an MLLM that demonstrates state-of-the-art performance in deepfake detection and reasoning. Future research could explore distilling reasoning annotations from a combination of different MLLMs to improve the robustness, diversity, and generalization of the generated explanations.

## 6. Conclusion

This work addresses the critical challenge of deepfake detection in the era of synthetic media, where explaining *why* an image is classified as real or fake is as important as the classification itself. We introduce a reasoning-annotated dataset, a multimodal architecture for deepfake detection and explainability, and Paragraph-level Relative Policy Optimization (PRPO), a reinforcement learning algorithm that enhances the reasoning capabilities of multimodal large language models (MLLMs) by aligning their explanations with visual evidence at a granular level. PRPO encourages models to generate detailed, evidence-grounded explanations without requiring extensive retraining on annotated data. Extensive experiments demonstrate that our approach substantially improves both detection accuracy and explanation faithfulness. PRPO paves the way for future research on integrating structured reasoning with vision-language models in safety-critical applications. To apply PRPO in broader vision-language reasoning tasks, such as visual question answering or visual entailment, future work could explore adapting the paragraph-level reward structure to domains that require structured, multi-paragraph reasoning, where appropriately designed rewards may guide models toward more coherent and evidence-aligned outputs.

## Impact Statement

The paper focuses on improving deepfake detection performance and explainability by introducing a new reasoning-annotated dataset DF-R5, a new multimodal detection model, DX-LLaVA, and a test-time reinforcement learning method, PRPO, that encourages alignment between visual evidence and generated reasoning. The intended benefit of this work is to support the development of reliable and interpretable deepfake detection systems. The system can assist human analysis in media forensics, media content verification, or digital authentication. By improving detection and explainability performance, the proposed method can reduce meaningless predictions in binary detection systems using opaque models and facilitate better human-AI collaboration.

As with many other deepfake detection systems, our work will be limited when facing new kinds of deepfakes in the future. Incorrect predictions in downstream tasks could also occur if appropriate safeguards are not deployed. In addition, generated reasoning may be misunderstood by non-expert analysts.

Finally, we emphasize that the proposed approach can be used as a decision-support tool, rather than as a sole decision-making system by itself. Beyond deepfake detection, future work may explore potential broader impact on other vision-language research tasks, such as visual question answering or visual entailment, where alignment between visual and textual information is considered important to reduce hallucinated answers.

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

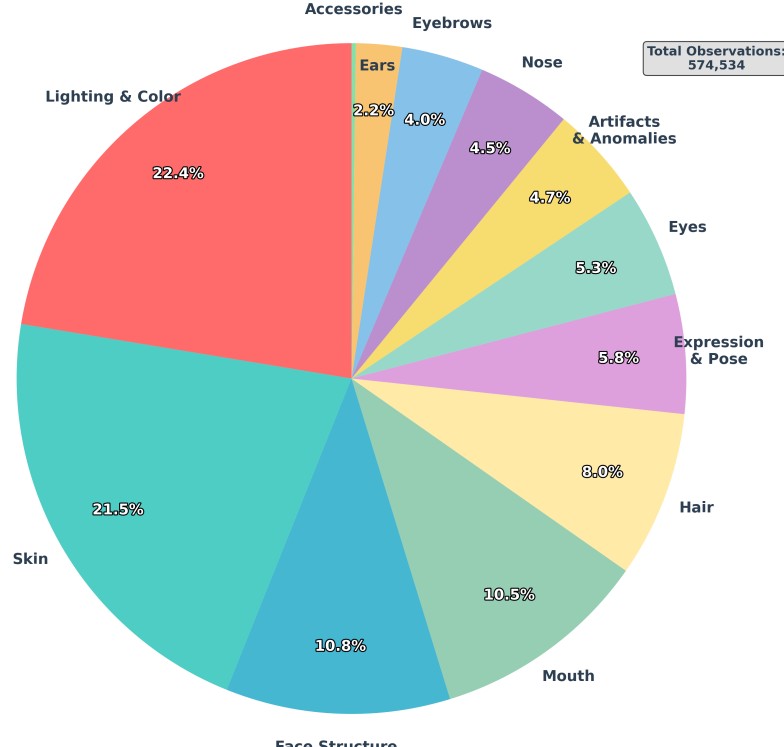

*Figure 4.* Distribution of deepfake detection features by category in the DF-R5 dataset (total of 574,534 feature observations), distilled from Gemini.

## A. Dataset construction and analysis

### A.1. Analyses on Deepfake Feature Collection

Figure 4 presents the distribution of deepfake-related features in our DF-R5 dataset, as distilled from Gemini's annotations. The dataset contains a total of 574,534 feature observations spanning diverse facial and contextual attributes. The most frequently mentioned categories are **Lighting & Color** (22.4%) and **Skin** (21.5%), which together account for nearly half of all annotations. This indicates that color tone mismatches, unnatural lighting, and irregular skin textures remain the most salient artifacts identified by Gemini. Other prominent categories include **Face Structure** (10.8%), **Mouth** (10.5%), and **Hair** (8.0%), which correspond to fine-grained facial details that are particularly sensitive to generative inconsistencies. Smaller proportions are attributed to features such as **Eyes** (5.3%), **Nose** (4.5%), **Eyebrows** (4.0%), and **Ears** (2.2%), as well as higher-level attributes like **Expression & Pose** (5.8%), **Artifacts & Anomalies** (4.7%), and **Accessories** (0.2%). These occur less frequently either due to their relatively small size in the image or the semantic complexity required to identify them. Overall, the distribution indicates that MLLMs capture a diverse range of deepfake characteristics, with **Lighting & Color** and **Skin** being the most prominent and error-prone regions in deepfake generation.

*Table 12.* List of 74 forensic-relevant features for deepfake detection.

| Index | Feature Name |
|-------|--------------|
| 1 | Inconsistent pupil shape, size, or symmetry. |
| 2 | Unnatural or missing eye specular highlights (catchlights). |
| 3 | Irregular or unnatural iris detail, pattern, or color. |
| 4 | Sclera (whites of eyes) with unnatural color, brightness, or texture. |
| 5 | Asymmetric or unnatural eyelid shape or creases. |
| 6 | Misaligned eye gaze direction. |
| 7 | Unnatural or blocky eyelashes. |
| 8 | Anomalies in eye structure (e.g., double irises/pupils, artificial tear ducts). |

Table 12 – continued from previous page

| Index | Feature Name |
|---|---|
| 9 | Unnatural skin texture (e.g., overly smooth, plastic-like, rough, lack of detail). |
| 10 | Inconsistent skin texture or detail across different facial regions. |
| 11 | Lack of realistic skin pores or inconsistent pore distribution. |
| 12 | Repetitive patterns in skin texture. |
| 13 | Unnatural or inconsistent skin tone or color patches. |
| 14 | Skin color mismatch between the face and neck, ears, or surrounding body. |
| 15 | Unnatural shininess, glossiness, or lack of expected specular highlights on skin surface. |
| 16 | Missing, unnatural, or misplaced blemishes, moles, scars, or freckles. |
| 17 | Unnatural or inconsistent wrinkles, folds, or creases. |
| 18 | Overexposed or underexposed skin patches. |
| 19 | Color banding or pixel noise in skin areas. |
| 20 | Lack of natural micro-variations in skin appearance. |
| 21 | Teeth with unnatural uniformity (shape, size, color, alignment, brightness). |
| 22 | Incorrect number or shape of visible teeth. |
| 23 | Teeth blending unnaturally into lips or gums, or unnatural gum line/spacing. |
| 24 | Pixelated, stretched, smudged, or artifact-laden teeth. |
| 25 | Unnatural lip contour, shape, or symmetry. |
| 26 | Unnatural lip color, texture, or color bleeding. |
| 27 | Sharp or unnatural corners of the mouth. |
| 28 | Unnatural transition between lips and teeth or inner mouth. |
| 29 | Unrealistic or missing tongue (if visible). |
| 30 | Misshapen philtrum (groove above upper lip). |
| 31 | Unnatural nose shape, proportions, or structural detail. |
| 32 | Asymmetric, smudged, or poorly defined nostrils. |
| 33 | Incorrect or missing shadows cast by the nose. |
| 34 | Overly smoothed nasal bridge. |
| 35 | Unnatural or asymmetric ear shapes or structures. |
| 36 | Ears inconsistent in size or position relative to the face. |
| 37 | Unnatural ear lobe attachment or blending. |
| 38 | Misaligned, asymmetric, or incomplete eyebrows. |
| 39 | Eyebrows blending unnaturally with skin or hair. |
| 40 | Unusual eyebrow thickness variation or shape. |
| 41 | Artificial, unnatural, sharp, or irregular hairline. |
| 42 | Unrealistic hair strand flow, shape, texture, or detail. |
| 43 | Hair blending unnaturally with the background or skin. |
| 44 | Artifacts, smudging, or unnatural uniformity in facial hair (beard/mustache/stubble). |
| 45 | Artificial blending or artifacts at hair roots. |
| 46 | Excessive or unnatural facial symmetry or asymmetry beyond natural variation. |
| 47 | Disproportionate facial features or overall distortion of facial structure/proportions. |
| 48 | Misaligned facial landmarks or features shifted off anatomical norms. |
| 49 | Lack of realistic depth or 3D appearance in facial structure. |
| 50 | Unnatural or overly defined cheekbone highlights or shadows. |
| 51 | Blurry, jagged, or wavy jawline edges or unnatural curvature. |
| 52 | Lack of definition in underlying bone or muscle structure relative to apparent age/body type. |
| 53 | Flat or unrealistic dimples. |
| 54 | Artificially thickened neck structure. |
| 55 | Inconsistent lighting direction or quality on different parts of the face or relative to the environment. |
| 56 | Shadows that contradict scene lighting, are missing, or unnaturally placed on the face. |
| 57 | Facial highlights (not specular) in incorrect positions or unnaturally placed. |
| 58 | Blurry, poorly defined, or overly sharp facial boundaries (face/neck, face/hair, face/background). |

Table 12 – continued from previous page

| Index | Feature Name |
|---|---|
| 59 | Visible blending artifacts, seams, ghosting, or glitch-like artifacts near facial edges or transitions. |
| 60 | Incorrect or missing reflections, warping, or distortion in glasses or other transparent objects near the face. |
| 61 | Missing, distorted, or misaligned jewelry, earrings, or other accessories. |
| 62 | Clothing textures blending unnaturally into facial skin or boundaries. |
| 63 | Background distortion, anomalies, or inconsistencies near the face or head. |
| 64 | Inconsistent image resolution, pixelation, or sharpness between the face and surroundings. |
| 65 | Inconsistent noise pattern or grain level between the face and rest of the image. |
| 66 | Overall color palette, white balance, or color fringing/halos inconsistent with the environment or rest of the image. |
| 67 | Repeating elements within features (not limited to skin texture). |
| 68 | Lack of realistic depth of field effects on facial elements. |
| 69 | General artifacts or visual noise not specific to a feature. |
| 70 | Unrealistic, frozen, rigid, or unnatural facial expressions. |
| 71 | Facial expression inconsistent with other features, context, or situation. |
| 72 | Unnatural stretching or distortion of features during apparent expression. |
| 73 | Facial pose or orientation inconsistencies. |
| 74 | Unnatural makeup patterns that appear digitally applied or inconsistent. |

## A.2. Prompts for Dataset Generation

In this section, we provide the full prompts used in our feature discovery (step 1), feature scoring (step 2), and reasoning generation processes (step 3). In step 1, we use the prompt in Figure 5 to generate $K = 50$ distinct visual characteristics from each of the four MLLMs (Gemini 2.5, GPT-4o, Qwen 2.5-Max, and LLaMA 4 Maverick). We then consolidate the $4 \times K = 200$ features into a unified list using the prompt in Figure 6. The final set of 74 consolidated features is reported in Table 12. In step 2, we use the prompt in Figure 7 to systematically score each of the $k = 74$ consolidated features for every image. Finally, in step 3, we use the prompt in Figure 8 to group the real-indicative features into logical categories and provide reasoning for each group.

---

**[Feature Discovery] Prompt for Multimodal LLMs**

Generate a list of {K} distinct and commonly observed visual characteristics that can help identify deepfake facial images.
Each characteristic should be:

- Clearly indicative of potential manipulation or digital forgery,

- Concise, unambiguous, and non-redundant,

- Focused on detectable artifacts, inconsistencies, or unnatural details in facial structure, texture, lighting, or surrounding context.

- Avoid repetition, each characteristic must describe a completely different phenomenon or cue.

Avoid general statements and ensure each characteristic highlights a unique visual cue that can be evaluated from a static image. List them in bullet or numbered format.

---

*Figure 5.* Prompt for generating a comprehensive set of visual cues to identify deepfake facial images, used across Gemini, GPT, LLaMA, and Qwen.

---

**[Feature Discovery] Feature Consolidation Prompt for Multimodal LLMs**

You are provided with a list of the top `{K}x4={4*K}` common forensic-relevant features used to detect forgery in facial images, as analyzed by state-of-the-art large language models, including GPT-4o, Gemini 2.5 Flash, Qwen 2.5-Max, and LLaMA 4 Maverick.

Your task is to:

1. Combine all `{K}x4={4*K}` features across these models into a single unified list.

2. Eliminate duplicate or overlapping features to ensure clarity and uniqueness.

3. Ensure each feature:

   - Is clearly defined and focused on detecting forgery in visual facial content.
   - Reflects diversity across models but avoids any redundancy.
   - Maintains precise and non-ambiguous language.

**Output format:**
A final list of unique and consolidated features, each on a separate line, numbered from 1 to N.
The provided features are:
**GPT-4o:** `{features_gpt}`
**Gemini 2.5:** `{features_gemini}`
**Qwen 2.5:** `{features_qwen}`
**LLaMA 4:** `{features_llama}`

---

*Figure 6.* Prompt for consolidating $4 \times K$ (e.g., $4 \times 50$) forensic-relevant features into a unified and non-redundant list, used across GPT-4o, Gemini 2.5, Qwen 2.5, and LLaMA 4.

## A.3. Paragraph Separation

For each image, we semantically partition the ground-truth reasoning according to the grouping results generated by Gemini. Each group is explicitly delineated using predefined markers. As illustrated in Fig. 9, the resulting annotations are formatted with paragraph-level separation. During PRPO-based test-time adaptation, rewards are applied at both the paragraph level and the final answer level.

## A.4. Reasoning diversity vs. Gemini reference

One concern that can be raised is whether our DX-LLaVA and PRPO closely mimic Gemini's reasoning style. To this end, we conduct a linguistic analysis by randomly sampling 1,000 images per domain (resulting in a total of 5,000 test images). For each image, we compare the generated reasoning from DX-LLaVA and PRPO with the ground truth from Gemini. In particular, we used $n \in \{1, 2, 3, 4\}$ to conduct a $n$-gram overlapping analysis. For each pair of generated reasoning and ground truth, we extract all consecutive $n$-word sequences and compute the percentage of overlapping sequences. The results are then averaged over 5,000 pairs to report corpus-level similarity measures. In addition, we further measure reasoning diversity by counting the number of unique visual features (which are distinct deepfake characteristics such as "blurry edges", "lighting inconsistencies", etc.) and creative features (which are features never seen in the Gemini corpus). The results are reported in Table 13. The results show that the reasoning generated by our DX-LLaVA and PRPO exhibits extremely low multi-gram similarity to Gemini's reference, especially for higher $n$-grams, with around 5.5% similarity for 3-grams and only 2.3% for 4-grams. Moreover, PRPO also produces richer deepfake characteristics with over 1,500 unique deepfake features and over 400 creative visual features, which are not present in Gemini's annotations. This analysis indicates the diversity of our corpus, which does not rely on or overfit to Gemini's reasoning style.

---

**[Feature Scoring] Feature Scoring Prompt for Gemini**

Given the attached image, evaluate each of the {k} listed deepfake characteristics. For each characteristic, respond with:

- Real (-1) if the characteristic appears natural,

- Fake (+1) if the characteristic clearly indicates digital forgery or manipulation,

- Uncertain (0) if the characteristic cannot be clearly evaluated from the provided image.

Provide your answers in the following format:

```
{feature 1: <score>},
{feature 2: <score>},
 ...
{feature k: <score>}.
```

Finally, based on your evaluation, provide your overall judgment clearly as:

```
Final Answer: <real/fake>
```

Note that your score of each feature should be fair, independent marking without bias.

---

*Figure 7.* Prompt for systematic scoring of $k = 74$ forensic-relevant characteristics in deepfake images, requiring per-feature evaluation and a final overall decision.

*Table 13.* Reasoning diversity against Gemini's reference across five domains.

| Metric | DX-LLaVA | PRPO |
|---|---|---|
| 1-gram overlap with Gemini (%) | 48.94 | 48.32 |
| 2-gram overlap with Gemini (%) | 15.89 | 16.05 |
| 3-gram overlap with Gemini (%) | 5.51 | 5.56 |
| 4-gram overlap with Gemini (%) | 2.36 | 2.35 |
| Unique visual features mentioned | 550 | **1,593** |
| Creative visual features | 20 | **404** |

## B. Reward analysis and diagnostics

### B.1. The effectiveness of Visual Consistency Reward (VCR)

To answer the question of how enhancing VCR captures fine-grained visual forensic cues, we analyze its sensitivity to pixel-level changes. We randomly sample 100 images per domain and apply common frequency-domain perturbations (Gaussian noise, brightness change, Gaussian blur, contrast adjustment, JPEG compression, and rotation). For each image, we use our trained model to generate 16 responses and compute the final prediction using the averaged majority-vote accuracy (Avg Acc). The results are shown in Table 14, which indicate that the generated answers become more varied when these perturbations are applied. VCR loses consensus among the 16 responses, especially under high-strength perturbations. In particular, Gaussian noise with strength $\sigma = 35$ and JPEG compression with compression quality $= 20$ cause significant performance drops (-27.75% and -30.72%, respectively). This demonstrates that VCR does in fact depend on pixel-level fine-grained forensic consistency rather than just high-level semantic alignment.

### B.2. Detailed Computation of Prediction Consistency Reward (PCR)

The Prediction Consistency Reward is computed through paragraph-level evidence scoring. Each paragraph is analyzed using dictionaries of real terms $\mathcal{R}$, fake terms $\mathcal{F}$, and negation terms $\mathcal{N}$ to determine whether it supports a "real" or "fake"

---

**[Reasoning Generation] Grouping and Reasoning Prompt for Deepfake Features**

Analyze the provided image, which has `{n_features}` features which indicate that the image is `{label}`. Your task is to first group these features into a maximum of `{M}` logical groups based on their conceptual similarity. Then, for each group, provide a concise reasoning that explains what the features within that group collectively suggest about the authenticity of this specific image. Instead of defining the features in general, describe what is notable or unusual (if the image is "fake") or typical (if the image is "real") about these features in the context of the image.

**The features indicating the image is `{label}` are:**

- `{feature 1}`

- `{feature 2}`

  `...`

- `{feature n_features}`

The ground truth label for this image is: `{label}`

**Please provide your analysis in JSON format following this exact structure:**

```
{
  "groups": {
    "group_name_1": ["feature_name_a", "feature_name_b", ...],
    "group_name_2": ["feature_name_c", "feature_name_d", ...],
    ...
  },
  "group_name_1": "reasoning for group 1",
  "group_name_2": "reasoning for group 2",
  ...,
  "answer": "ground truth label"
}
```

---

*Figure 8.* Prompt for grouping real-indicative features into logical categories with reasoning. Each image has different variables (e.g., `n_features`, `M`, `label`, and the list of features).

label. Scores are accumulated based on the frequency of matched terms, and a label is assigned accordingly. All paragraphs contribute to a majority vote, producing the majority label $a_{\text{maj}}$. The reward is set high when the majority label matches the final answer $a_{\text{final}}$, and reduced when inconsistencies are detected. The detailed procedures are presented in Algorithms 1, 2, and 3. This reward design enforces consistency between the final answer and the majority of paragraph-level predictions, thereby improving the reliability of the model's outputs.

## C. Architecture and representation ablations

### C.1. DX-LLaVA architecture details

In this section, we provide the detailed architecture and training configuration of DX-LLaVA in Table 15. The CLIP ConvNeXT vision encoder was used, accounting for 2.82% of the total model parameters, and was kept frozen during training. The classifier $\mathcal{C}(\cdot; \phi)$ is also a lightweight 2-layer MLP, contributing only 0.03% of the model size . All trainable layers are optimized using AdamW (Loshchilov & Hutter, 2019) with a learning rate of $2 \times 10^{-5}$. The training pipeline of DX-LLaVA is built upon the open-source LLaVA codebase [2].

---

[2]https://github.com/haotian-liu/LLaVA.git

<**Group 1**> **Skin Texture Anomalies**
The skin exhibits an unnaturally smooth appearance...

<**Group 2**> **Lighting and Shadow Inconsistencies**
The lighting on the face is inconsistent...

**### Answer:** fake

*Figure 9.* Example of grouped visual forensic reasoning used in dataset annotation.

*Table 14.* Effect of pixel-level perturbations on VCR majority-vote accuracy.

| Perturbation | Factor | VCR (Avg Acc %) | Change (%) |
|---|---|---|---|
| Unperturbed | – | **96.00** | – |
| Gaussian Noise | $\sigma = 35$ | 68.25 | $-27.75$ |
| Gaussian Blur | kernel $= 9$, $\sigma = 3$ | 79.00 | $-17.00$ |
| Brightness | factor $= 1.4$ | 95.13 | $-0.87$ |
| Contrast | factor $= 1.5$ | 91.19 | $-4.81$ |
| JPEG Compression | quality $= 20$ | 65.28 | $-30.72$ |
| Rotation | degrees $= 5$ | 82.75 | $-13.25$ |

## C.2. ConvNeXT vs. ViT Backbone Comparison

### C.2.1. SALIENCY MAP COMPARISON.

To better understand the representational differences between CLIP ConvNeXT and CLIP ViT backbones in deepfake detection, we analyze their saliency behaviors (Simonyan et al., 2014) using the visualizations in Figure 10. Although both models originate from CLIP (Radford et al., 2021) and are fine-tuned on the same deepfake detection task, their attention patterns diverge significantly. In the middle-left image, the CLIP ConvNeXT backbone produces sharply localized saliency concentrated around key facial regions, achieving a high prediction confidence (99.3%). In contrast, the middle-right image shows that the CLIP ViT backbone yields more diffuse and spatially scattered responses with noticeably lower confidence (47.2%). As highlighted in the rightmost column of Figure 10, the largest discrepancies cluster around the central facial area, suggesting that CLIP ConvNeXT attends more strongly to discriminative forensic cues that are critical for deepfake detection.

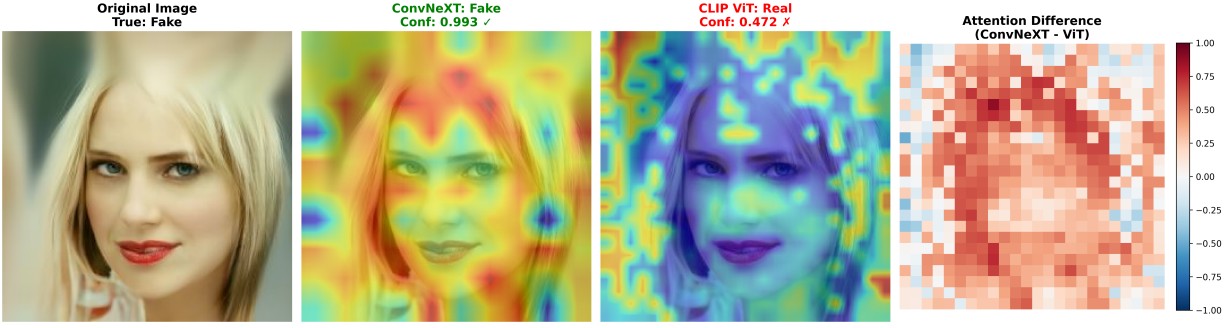

*Figure 10.* Saliency map comparison between CLIP ConvNeXT and CLIP ViT on a fake image.

### C.2.2. FEATURE VISUALIZATIONS AND CONFIDENCE DISTRIBUTIONS.

We compare the learned feature representations of the two backbones in both **intra-domain** and **inter-domain** settings to better understand their effectiveness in deepfake detection. As illustrated in Figures 11 and 12, CLIP ConvNeXT produces well-separated clusters for real (blue) and fake (red) samples, whereas CLIP ViT exhibits substantial overlap between the two classes. The corresponding confidence distributions further highlight CLIP ConvNeXT's superior calibration: its predictions form clear peaks near 0 (real) and 1 (fake), making them easily separable even with a linear classifier. In contrast, CLIP ViT's outputs concentrate around the decision boundary, reflecting higher uncertainty. These observations collectively demonstrate

---

**Algorithm 1** Prediction Consistency Reward Computation

---

1: **Input:** paragraph index $i$, paragraph $p_i$, all paragraphs $\mathcal{P}$ in $L$ samples, final answer $a_{\text{final}}$, number of paragraphs $V$, majority answer $a_{\text{maj}}$, prediction consistency reward $r_i$
2: Compute paragraph scores: $u_i \leftarrow$ score_paragraph$(p_i)$
3: Predict label: $\hat{y}_i \leftarrow$ predict_label$(u_i)$
4: **if** $p_i$ is the final answer paragraph **then**
5:     Initialize: $V_{\text{real}} \leftarrow 0$, $V_{\text{fake}} \leftarrow 0$
6:     **for** $j = 1$ to $i - 1$ **do**
7:         $u_j \leftarrow$ score_paragraph$(p_j)$
8:         $\hat{y}_j \leftarrow$ predict_label$(u_j)$
9:         **if** $\hat{y}_j =$ "real" **then**
10:            $V_{\text{real}} \leftarrow V_{\text{real}} + 1$
11:         **else if** $\hat{y}_j =$ "fake" **then**
12:            $V_{\text{fake}} \leftarrow V_{\text{fake}} + 1$
13:         **end if**
14:     **end for**
15:     **if** $V_{\text{real}} > V_{\text{fake}}$ **then**
16:         $a_{\text{maj}} \leftarrow$ "real"
17:     **else if** $V_{\text{fake}} > V_{\text{real}}$ **then**
18:         $a_{\text{maj}} \leftarrow$ "fake"
19:     **else**
20:         $a_{\text{maj}} \leftarrow a_{\text{final}}$
21:     **end if**
22:     **if** $a_{\text{maj}} = a_{\text{final}}$ **then**
23:         $r_i \leftarrow 1.0$
24:     **else**
25:         $r_i \leftarrow 0.0$
26:     **end if**
27: **else**
28:     $r_i \leftarrow 1.0$
29: **end if**
30: **Return:** $r_i$

---

**Algorithm 2** Paragraph Scoring Function (score_paragraph)

---

1: **Input:** Paragraph text $p$
2: **Initialize:** $s_{real} \leftarrow 0.0$, $s_{fake} \leftarrow 0.0$
3: Declare real patterns $\mathcal{R}$, fake patterns $\mathcal{F}$, negation patterns $\mathcal{N}$
4: **for** each real term match in $p$ **do**
5:     **if** a negated term exists **then**
6:         $s_{fake} \leftarrow s_{fake} + 1$
7:     **else**
8:         $s_{real} \leftarrow s_{real} + 1$
9:     **end if**
10: **end for**
11: **for** each fake term match in $p$ **do**
12:     **if** a negated term exists **then**
13:         $s_{real} \leftarrow s_{real} + 1$
14:     **else**
15:         $s_{fake} \leftarrow s_{fake} + 1$
16:     **end if**
17: **end for**
18: **Return:** $\{s_{real}, s_{fake}\}$

---

---

**Algorithm 3** Predict Label from Paragraph Scoring (predict_label)

---

**Require:** Paragraph text $p$
**Ensure:** Predicted label $\ell \in \{$"real", "fake"$\}$
1: **if** $s_{real} \geq s_{fake}$ **then**
2:     $\ell \leftarrow$ "real"
3: **else**
4:     $\ell \leftarrow$ "fake"
5: **end if**
6: **Return:** $\ell$

---

CLIP ConvNeXT's stronger ability to capture discriminative forensic cues and make confident deepfake detection decisions.

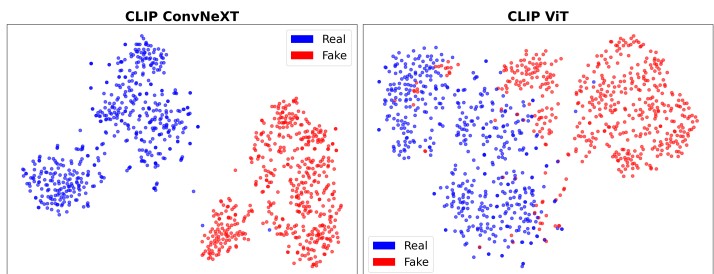

*(a)* $t$-SNE visualization of feature embeddings

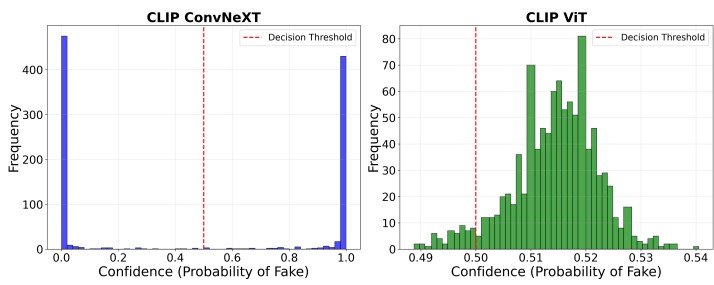

*(b)* Confidence score distribution

*Figure 11.* Intra-domain analysis comparing ConvNeXT and ViT backbones.

## D. Reasoning quality evaluation

### D.1. Human Evaluation

In this experiment, we conducted a human-evaluation study with **8** participants, split into a *lower-expertise* group (4 researchers with general computer vision experience) and an *expert* group (4 researchers with specialized expertise in deepfake analysis). Each participant rated explanations from anonymous models on 5 randomly sampled images using the same rubric as GPT-4o. The results in Table 16 show that human judgments strongly align with GPT-4o's assessments, and both consistently rank PRPO highest across all evaluation criteria. The evaluation criteria (CAC, EGIA, RQ, CC, and CU) are presented in Section 4.4. This cross-validation confirms that GPT-4o provides reliable and unbiased evaluations of explanation quality.

### D.2. Prompt for Qualitative Evaluation

We provide the full prompt used to evaluate the quality of reasoning responses from different models in Figure 13.

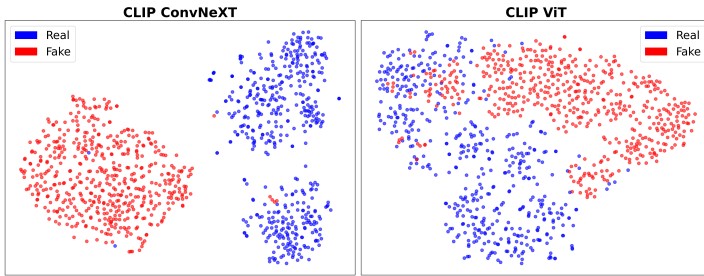

*(a)* $t$-SNE visualization of feature embeddings across domains

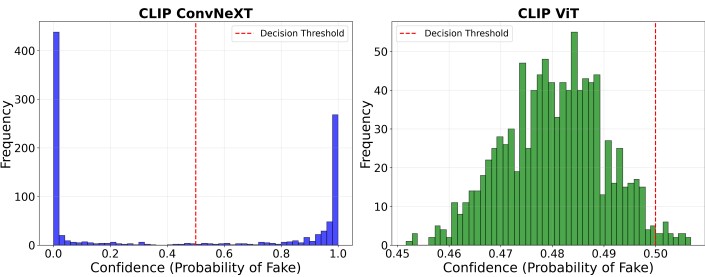

*(b)* Confidence score distribution across domains

*Figure 12.* Inter-domain generalization analysis comparing ConvNeXT and ViT backbones on unseen **DDIM** domain.

*Table 15.* DX-LLaVA architecture and training configuration.

| Component | Parameters | Trainable | Learning Rate |
|---|---|---|---|
| CLIP ConvNeXT (vision encoder) | 196.2M | Frozen | - |
| Projection layer $W$ (2-layer MLP) | 23.1M | ✓ | $2 \times 10^{-5}$ |
| Vicuna-7B (language model) | 6.738B | ✓ | $2 \times 10^{-5}$ |
| Classifier $\mathcal{C}(\cdot; \phi)$ (2-layer MLP) | 2.1M | ✓ | $2 \times 10^{-5}$ |
| **Total trainable** | **6.763B** | - | - |

## D.3. Qualitative Analysis of Model Reasoning

In this section, we present qualitative comparisons of reasoning outputs from five vision–language models: Qwen2.5-VL-32B (Qwen et al., 2025), Gemma-3-27B (Team et al., 2025b), Gemini-2.5 (Team et al., 2025a), DX-LLaVA, and PRPO. We analyze three representative cases: Example 1 (fake), Example 2 (fake), and Example 3 (real), as shown in Figures 14, 15, and 16.

The qualitative results highlight clear differences in reasoning quality among the models. General-purpose MLLMs often provide generic or surface-level observations, while our proposed models (DX-LLaVA and PRPO) deliver more detailed and forensic-oriented explanations. They consistently identify concrete artifacts such as abnormal skin texture, inconsistent lighting, and distorted accessories, and they organize their findings into well-structured, category-based analyses. In addition, they show reliable performance on both fake and real images, offering explanations that are more thorough, interpretable, and aligned with key forensic cues.

## E. Additional ablation studies

### E.1. Sensitivity analysis for hyperparameters $\alpha$ and $\beta$.

We conduct an ablation study to testify the sensitivity of $\alpha$ and $\beta$. The results are reported in Table 18 and Table 19. Overall, the performance remains stable across different ranges of values, with above 95% F1 for $\alpha$ and above 93% F1 for $\beta$. The increasing $\alpha$ leads to improvement of precision (99.80%), and a slight drop in recall. While increasing $\beta$ tends to slightly improve precision (up to 99.78%) but can reduce recall (93.6%). Depending on requirements of application, $\alpha$ and $\beta$ can be

*Table 16.* Human vs. GPT-4o evaluation.

| Group | Model | CAC | EGIA | RQ | CC | CU | Average |
|---|---|---|---|---|---|---|---|
| All Human Evaluators | Gemini-2.5 | 4.225 | 4.350 | 4.275 | 3.950 | 4.300 | 4.220 |
| | **PRPO** | **5.00** | **4.625** | **4.475** | **4.575** | **4.475** | **4.630** |
| GPT-4o | Gemini-2.5 | 3.98 | 4.16 | 4.23 | 4.05 | 4.60 | 4.20 |
| | **PRPO** | **4.42** | **4.56** | **4.58** | **4.50** | **4.69** | **4.55** |

*Table 17.* Comprehensive detection performance (%) of our method compared with deepfake detection baselines across five domains.

| Method | → DDIM | | | | → PixArt | | | | → SD2.1 | | | |
|---|---|---|---|---|---|---|---|---|---|---|---|---|
| | Acc | Prec | Rec | F1 | Acc | Prec | Rec | F1 | Acc | Prec | Rec | F1 |
| LLaVA | 63.30 | 88.78 | 34.67 | 49.86 | 70.90 | 91.86 | 50.84 | 65.46 | 53.30 | 80.00 | 15.91 | 26.54 |
| DE-FAKE | 46.30 | 40.63 | 4.95 | 8.83 | 86.30 | 91.42 | 81.99 | 86.45 | 95.40 | 92.43 | 99.43 | 95.80 |
| FakeShield | 44.51 | 35.66 | 44.51 | 31.84 | 88.70 | 89.59 | 88.70 | 88.57 | 92.30 | 92.48 | 92.30 | 92.28 |
| UnivCLIP | 77.61 | 86.88 | 80.63 | 74.85 | 82.20 | 93.31 | 74.09 | 89.31 | 76.70 | 88.39 | 78.81 | 74.81 |
| SIDA | 71.46 | 79.34 | 72.66 | 70.07 | 68.00 | 65.41 | 84.80 | 73.86 | 92.41 | 92.42 | 92.33 | 92.37 |
| DX-LLaVA (ours) | 92.60 | 99.11 | 86.43 | 92.34 | 84.60 | 100.00 | 71.11 | 83.11 | 89.70 | 99.53 | 81.06 | 89.35 |
| PRPO (ours) | 95.80 | 98.79 | 93.14 | 95.88 | 88.60 | 99.29 | 79.17 | 88.10 | 94.80 | 96.67 | 93.37 | 94.99 |

| Method | → SiT | | | | → StyleGAN3 | | | | Average | | | |
|---|---|---|---|---|---|---|---|---|---|---|---|---|
| | Acc | Prec | Rec | F1 | Acc | Prec | Rec | F1 | Acc | Prec | Rec | F1 |
| LLaVA | 50.90 | 64.71 | 8.71 | 15.36 | 67.10 | 88.93 | 41.97 | 57.03 | 61.10 | 82.86 | 30.42 | 42.85 |
| DE-FAKE | 49.70 | 54.55 | 2.38 | 4.55 | 79.60 | 94.58 | 64.22 | 76.50 | 71.46 | 74.72 | 50.59 | 54.43 |
| FakeShield | 49.70 | 75.05 | 49.70 | 33.22 | 98.70 | 98.72 | 98.70 | 98.70 | 74.78 | 78.30 | 74.78 | 68.92 |
| UnivCLIP | 61.31 | 71.53 | 83.03 | 40.01 | 81.61 | 92.45 | 76.39 | 86.46 | 75.89 | 86.51 | 78.59 | 73.09 |
| SIDA | 56.29 | 76.55 | 56.72 | 46.53 | 95.01 | 95.19 | 94.91 | 94.98 | 76.63 | 81.78 | 80.28 | 75.56 |
| DX-LLaVA (ours) | 57.20 | 100.0 | 15.25 | 26.46 | 99.10 | 99.22 | 99.03 | 99.13 | 84.64 | 99.57 | 70.58 | 78.08 |
| PRPO (ours) | 66.60 | 63.01 | 81.98 | 71.26 | 99.30 | 99.23 | 99.42 | 99.32 | 89.02 | 91.40 | 89.42 | 89.91 |

adjusted to find optimal balance between minimizing false negatives and false positives.

### E.2. Computational Efficiency and Training Overhead

To investigate the computational cost of PRPO during test-time adaptation, we profile the model over the first 10 iterations and report the averaged measurements. Experiments are conducted on 8 NVIDIA H200 GPUs. Input images are resized $320 \times 320$, and the maximum generation length is 1024 tokens. We vary the batch size (BS) from 8 to 32 and measure metrics including average iteration time, standard deviation of iteration time, average throughput, standard deviation of throughput, average latency, and standard deviation of latency. The results are summarized in Table 20. Metrics show efficiency trend when increasing the batch sizes. For instance, the averaged throughput improves 70% from 0.0810 to 0.1379, while the averaged latency is reduced 46% from 13.8 seconds to only 7.4 seconds. This is expected because when the GPUs process more data, the per-sample cost consistently decreases. The variability during optimization also improves, where the standard deviation of latency drops from 4074.7 ms to 1168.1 ms, showing more stable optimization dynamics at larger batch sizes.

Table 21 further breaks down the computational cost into different components, including sampling, reward computation, actor update (Act. Update), reference policy (Ref. Policy) computation, and advantage (Adv.) calculation. It is noticeable that the sampling process accounts for the majority of the computational cost, at 89-90% across all batch sizes, while reward computation and actor/reference policy updates remain lightweight, with less than 6%. In particular, the cost of computing advantages is negligible. This again confirms that PRPO is efficient in computing and optimizing its rewards, as the majority of the cost comes from the sampling process, which is expected for GRPO-based reinforcement learning methods that rely on large language models to generate multiple groups of answers. Overall, larger batch sizes offer significantly more

*Table 18.* Sensitivity analysis of $\alpha$ on the unseen DDIM domain.

| $\alpha$ | Precision (%) | Recall (%) | F1 (%) |
|---|---|---|---|
| 0.0 | 98.97 | 91.62 | 95.15 |
| 1.0 | 99.20 | 94.86 | 96.98 |
| 10.0 | 98.79 | 93.14 | 95.88 |
| 20.0 | 99.80 | 93.33 | 96.46 |

*Table 19.* Sensitivity analysis of $\beta$ on the unseen DDIM domain.

| $\beta$ | Precision (%) | Recall (%) | F1 (%) |
|---|---|---|---|
| 0.0 | 99.57 | 88.95 | 94.00 |
| 0.001 | 99.58 | 89.33 | 94.20 |
| 0.005 | 99.57 | 88.57 | 93.80 |
| 0.01 | 99.57 | 88.19 | 93.60 |
| 0.05 | 99.57 | 89.14 | 94.10 |
| 0.1 | 99.78 | 88.00 | 93.60 |

*Table 20.* PRPO test-time adaptation performance across batch sizes.

| BS | Avg Iter Time (s) | Std Iter Time (s) | Avg Throughput (samples/s) | Std Throughput (samples/s) | Avg Latency (ms) | Std Latency (ms) |
|---|---|---|---|---|---|---|
| 8 | 110.7902 | 32.5974 | 0.0810 | 0.0311 | 13848.78 | 4074.68 |
| 16 | 153.0948 | 45.6196 | 0.1131 | 0.0294 | 9568.43 | 2851.22 |
| 32 | 237.6357 | 37.3804 | 0.1379 | 0.0211 | 7426.12 | 1168.14 |

computationally efficient and stable PRPO test-time adaptation.

*Table 21.* Component breakdown of PRPO computational cost.

| BS | Sampling (s) | Sampling (%) | Reward (s) | Reward (%) | Act. Update (s) | Act. Update (%) | Ref. Policy (s) | Ref. Policy (%) | Adv. (s) | Adv. (%) |
|---|---|---|---|---|---|---|---|---|---|---|
| 8 | 95.3877 | 89.08% | 3.4758 | 3.25% | 6.3794 | 5.96% | 1.8177 | 1.70% | 0.0222 | 0.02% |
| 16 | 131.0687 | 89.43% | 6.3492 | 4.33% | 7.1801 | 4.90% | 1.9357 | 1.32% | 0.0257 | 0.02% |
| 32 | 203.7870 | 90.14% | 11.6396 | 5.15% | 8.3056 | 3.67% | 2.2918 | 1.01% | 0.0550 | 0.02% |

### E.3. Intra-domain vs. Inter-domain Experimental Setup.

In Table 2, we compare two data-splitting strategies on the baseline LLaVA model. For the intra-domain setting, we randomly split our full dataset into train/validation/test with a ratio of 98%/1%/1%. All splits contain mixed samples from all five domains. For the inter-domain setting, we adopted a leave-one-domain-out protocol. We train on four domains (e.g., PixArt, SD, SiT, StyleGAN3) and evaluate on the held-out domain (e.g., DDIM). In Table 3, we evaluate the effect of adding the binary classification loss $\mathcal{L}_{\text{binary}}$ to the language modeling loss $\mathcal{L}_{\text{lm}}$ under the *inter-domain* setting.

### E.4. Examples of Success and Failure Cases.

In this section, we present qualitative examples of PRPO's reasoning compared to other models, and analyze a common failure mode in the SiT domain.

**Success case (grounded correction of hallucination).** The example in Figure 16 depicts a real image of a naturally photographed woman. The comparison between PRPO and MLLMs is described in Table 22.

The results show that Qwen and Gemma hallucinate generic artifact patterns ("overly smooth skin", "unnaturally perfect teeth") that fire on this real image because the CLIP similarity of these generic phrases to a real face is uniformly moderate; they fail to distinguish this specific image. PRPO's VCR reward penalized such descriptions during training, pushing the model toward image-grounded observations. The difference between "overly smooth" (generic) and "natural blemishes and creases in skin texture" (specific) is precisely what PRPO learns to discriminate through relative advantage scoring.

*Table 22.* Qualitative comparison of reasoning strategies on a real image (Figure 16).

| Model | Reasoning strategy | Prediction | Correct? |
|---|---|---|---|
| Qwen2.5-VL-32B | "Shiny, idealized skin, unnaturally perfect teeth, lighting lacks natural imperfections" | Fake | ✗ |
| Gemma-3-27B | "Overly smooth skin texture, slight asymmetry ears feels unnatural and poorly integrated with the head" | Fake | ✗ |
| DX-LLaVA | "Eye characteristics exhibit typical natural variations; skin tone consistent across regions; mouth shows natural contours; hair blends naturally with background" | Real | ✓ |
| **PRPO** | **"Natural catchlights present in eyes; natural blemishes and creases in skin texture; well-defined nostrils with natural shadow; hair exhibits realistic strand flow"** | **Real** | ✓ |

**Failure mode (sparse-artifact diffusion in SiT domain, F1 = 71.26%).** SiT-XL/2 (Ahmed et al., 2021) is our hardest domain and accounts for the majority of PRPO's remaining errors. The failure pattern is: (i) SiT images are perceptually near-perfect. Diffusion score matching produces globally consistent statistics; (ii) PRPO generates paragraphs describing real visual characteristics (natural texture, consistent lighting) that are *genuinely present* and receive high VCR (correctly aligned with the image); (iii) The majority vote across paragraphs leans toward "real" because authentic-looking features outnumber detectable artifacts (4 of 7 paragraphs describe real traits); and (iv) PRPO produces a confident "real" prediction that is incorrect.

Generally, the failure is that the absence of artifacts is itself evidence of a high-quality fake, which requires negative reasoning that our current reward structure does not explicitly incentivize. The same pattern does not occur for GAN-based fakes (StyleGAN, F1 = 99.32%) or DDIM (F1 = 95.88%), which have local spectral artifacts that VCR successfully grounds.

---

**Evaluation Prompt**

You are an expert evaluator for deepfake detection responses. Your task is to evaluate a given response to an image across five critical dimensions for deepfake detection accuracy and reliability. **Scoring Scale:** For each dimension, assign an integer score from **0 to 5**:

- 0 = Very poor / completely incorrect
- 1 = Poor
- 2 = Fair
- 3 = Good
- 4 = Very good
- 5 = Excellent

**Evaluation Dimensions**

1. **Classification Accuracy & Consistency:**
   Does the response correctly classify the image as real or fake?
   Is the classification consistent with both the ground truth and the reasoning provided?
2. **Reasoning Quality:**
   Does the response provide a logical, step-by-step explanation of its decision?
   Is the reasoning free from contradictions or irrelevant details?
3. **Evidence Grounding & Image Alignment:**
   Does the response cite specific visual artifacts that are actually present in the image?
   Does it avoid hallucinations (mentioning features not visible)?
4. **Confidence Calibration:**
   Is the expressed confidence level appropriate given the clarity of evidence in the image?
   Does the response avoid overstating or understating certainty?
5. **Clarity & Usefulness:**
   Is the response clear, well-structured, and easy to understand?
   Would it be useful for a human investigator verifying deepfake authenticity?

**Output Format**

Respond strictly in JSON with this structure:

```
{
  "classification_accuracy": <0-5>,
  "evidence_grounding": <0-5>,
  "reasoning_quality": <0-5>,
  "confidence_calibration": <0-5>,
  "clarity_usefulness": <0-5>,
  "justification": "<concise explanation of the scoring
  rationale>"
}
```

**Evaluation Task**

Now evaluate the given image with the following details:

**Response:** {response}
**Prediction:** {prediction}
**Ground Truth:** {ground_truth}

*Figure 13.* Prompt provided to evaluators for scoring deepfake detection responses on a 0-5 scale across five criteria.

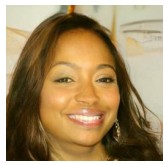

**Ground Truth:** Fake

**Question:**

*Analyze this image for any signs of digital manipulation or artificial generation and determine if it is real or fake. Provide your reasoning first, then conclude with a final answer.*

**Qwen2.5-VL-32B:**

The image appears to have natural lighting and shadows, which is consistent with authentic photographs. The details in the subject's facial features, hair, and clothing appear realistic and consistent with high-resolution photography. The background includes elements like an award or certificate, which seem to be part of a real event setting, adding context that supports authenticity. There are no visible distortions or anomalies typical of deepfake or digital manipulation, such as inconsistencies in texture, lighting, or facial features. The overall composition and framing do not exhibit sharp transitions or artifacts that would suggest artificial generation.
**Answer:** real

**Gemma-3-27B:**

The image exhibits some subtle inconsistencies in the specular highlights on the teeth, which could indicate minor smoothing or manipulation. However, these are not pronounced enough to definitively claim the image is fake. The skin texture, while appearing somewhat smoothed, isn't overly artificial. There are visible pores and subtle imperfections. The background appears slightly blurry and out of focus, which is common in candid photographs. It doesn't contain any obvious anomalies. The overall lighting and shadow play are plausible for a naturally lit scene. The quality of the image is not extremely high, which makes detecting minute artifacts more difficult, but also supports a more likely origin as a standard photograph.
**Answer:** real

**Gemini-2.5:**

The overall image has a highly artificial, airbrushed, and almost plastic-like quality that is common in AI-generated portraits. The skin texture appears unusually smooth and lacks the natural blemishes or variations typically found in real photographs. The background is very blurry and indistinct, a common characteristic of GAN-generated images where the focus is solely on the central subject. Some areas, like the hair near the top of the head, show slight pixelation or unnatural blending indicative of generation artifacts. The earring on the right side of the image (viewer's left) appears somewhat distorted and lacks clear definition, which is often a tell-tale sign of AI generation struggling with intricate details. The teeth appear unnaturally perfect and uniformly white, a common artifact in AI-generated faces striving for an 'ideal' appearance. The lighting seems somewhat flat and doesn't cast natural shadows or highlights consistently across the face. For instance, the light source creating reflections in the eyes isn't clearly defined.
**Answer:** fake

**DX-LLaVA (Ours):**

**Skin Texture and Detail Anomalies:** The skin in the image exhibits an unnatural smoothness and lack of fine detail, appearing overly uniform and plastic-like. There are inconsistencies in texture across different areas of the face, and the expected presence and distribution of realistic skin pores are absent or irregular. This suggests a lack of natural skin micro-variations typically found in real images.
**Skin Tone and Color Inconsistencies:** The skin tone appears unnatural and shows inconsistent color patches, which is not typical of natural skin pigmentation and lighting conditions.
**Teeth Uniformity and Appearance Issues:** The teeth display an unnatural degree of uniformity in terms of shape, size, color, alignment, and brightness. They also show signs of pixelation, stretching, smudging, or other artifacts, indicating manipulation.
**Teeth and Mouth Transition Anomalies:** The transition between the teeth, lips, and gums is unnatural, with the teeth appearing to blend unnaturally into the surrounding tissues or exhibiting an irregular gum line and spacing. The transition between the lips and the inner mouth also appears artificial.
**Lip Appearance and Transition Issues:** The lips have an unnatural contour, shape, or symmetry, and their color and texture appear artificial, with potential color bleeding. The transition between the lips and the teeth or inner mouth is also unnatural.
**Facial Boundary and Blending Artifacts:** The boundaries of the face, particularly where it meets the neck, hair, or background, are either unnaturally blurry, poorly defined, or excessively sharp. There are visible blending artifacts, seams, ghosting, or glitch-like anomalies near these transitions, suggesting the face has been composited or altered.
**Overall Image Consistency Issues:** The image exhibits inconsistencies in resolution, pixelation, or sharpness between the facial area and the surrounding environment. Additionally, there are general artifacts or visual noise present throughout the image that are not tied to specific features, indicating potential manipulation or generation processes.
**Answer:** fake

**PRPO (Ours):**

**Skin Texture and Detail Anomalies:** The skin in the image exhibits an unnatural smoothness and lacks the expected variations in texture, pores, and micro-details found in real skin. Different areas of the face show inconsistencies in how these details are rendered, suggesting artificial generation or manipulation rather than a natural photographic capture.
**Skin Tone and Color Inconsistencies:** The skin tone appears unnatural and inconsistent in patches, lacking the subtle gradients and natural color variations typically seen in human skin, indicating a potential manipulation that has failed to accurately replicate realistic skin pigmentation.
**Skin Surface Properties:** The skin surface shows an unnatural shininess or glossiness, and the expected specular highlights that would naturally occur under the apparent lighting conditions are either absent or incorrectly rendered, suggesting an artificial surface property that doesn't react realistically to light.
**Teeth Uniformity and Appearance:** The teeth display an unnatural degree of uniformity in shape, size, color, alignment, and brightness. Furthermore, they show signs of pixelation, stretching, smudging, or other artifacts, which are highly indicative of digital manipulation rather than natural dental features captured in an image.
**Teeth and Mouth Transition Issues:** The transition between the teeth and the surrounding lips and gums is unnatural. The teeth appear to blend unnaturally into the lips or gums, and the gum line or spacing is unrealistic. The transition also appears to be pixelated, stretched, or smudged, suggesting a failure in seamlessly integrating the teeth into the surrounding mouth area.
**Lip Appearance Anomalies:** The lips exhibit unnatural color, texture, or color bleeding, which is not typical of natural lip appearance. This suggests that the lip area has been artificially altered or generated, resulting in an unconvincing appearance.
**General Image Artifacts:** The presence of general artifacts or visual noise that are not tied to a specific facial feature suggests overall image degradation or the presence of residual artifacts from a generative process, further contributing to the assessment of the image as fake.
**Answer:** fake

*Figure 14.* Qualitative comparison of model reasoning on Example 1.

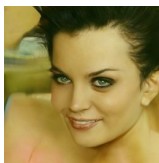

**Ground Truth:** Fake

**Question:**

*Analyze this image for any signs of digital manipulation or artificial generation and determine if it is real or fake. Provide your reasoning first, then conclude with a final answer.*

**Qwen2.5-VL-32B:**

The image appears to have overly smooth and perfect skin, which is a common indicator of digital manipulation or artificial enhancement. The lighting seems consistent and lacks natural variations, suggesting a filter or post-processing effect rather than natural light sources. The subject's features, such as the eyebrows and lips, are highly defined and symmetrical, which is typical in AI-generated or heavily retouched images. There is a lack of visible pores, blemishes, or natural imperfections, which are usually present in real photographs. While these features suggest manipulation, it is not entirely conclusive since high-quality editing can replicate realistic textures.
**Answer:** fake

**Gemma-3-27B:**

The image exhibits inconsistencies and artifacts strongly indicative of AI generation. Specifically, there's a blurring and waxiness to the skin texture that isn't consistent with natural skin. The eyes are unnaturally sharp and detailed in comparison to the surrounding skin, creating a jarring contrast. There's a slight asymmetry between the left and right sides of the face, but it doesn't feel natural asymmetry, rather like a model wasn't quite aligned during rendering. The lighting is somewhat inconsistent, particularly around the lips and the slight shadowing on the nose. The overall aesthetic and hyperrealism are characteristic of AI-generated portraits, like those produced by Stable Diffusion or Midjourney.
**Answer:** fake

**Gemini-2.5:**

The overall image quality and texture exhibit characteristics often seen in AI-generated images, such as a slightly artificial smoothness and lack of fine detail that would normally be present in a photograph. The hair at the top of the head appears somewhat indistinct and merges into the background in a way that is unnatural, lacking individual strands or clear definition. The background is very blurry and amorphous, which is a common characteristic of AI-generated portraits attempting to focus on the subject while creating a non-distracting backdrop. While the facial features are generally coherent, there's a subtle 'uncanny valley' effect, where something feels slightly off despite appearing human-like. The lighting seems a bit flat or artificially applied, not quite conforming to a natural light source that would create more varied shadows and highlights. There are some slight distortions or inconsistencies in the contours around the face and possibly the ears, which are common artifacts in GAN-generated faces.
**Answer:** fake

**DX-LLaVA (Ours):**

**Eye Anomalies:** The eyes in the image exhibit multiple signs of manipulation, including unnatural pupil characteristics, missing or artificial highlights, irregular iris details, and unusual sclera appearance. The eyelids and eyelashes also show unnatural shapes and textures, and the gaze direction appears misaligned, collectively indicating that the eyes were likely digitally altered or generated.
**Skin and Texture Irregularities:** The skin in the image lacks realistic detail and texture, appearing overly smooth or inconsistent across different areas. There is a noticeable absence of natural pores and micro-variations, along with unnatural color patches and shininess. The presence of color banding or pixel noise further suggests digital manipulation of the skin surface.
**Mouth and Dental Abnormalities:** The mouth and teeth display significant irregularities. The teeth show unnatural uniformity and blending issues with the lips and gums, along with pixelation and artifacts. The lips themselves have unnatural contours, colors, and transitions, and the corners of the mouth appear sharp or unnatural, pointing to digital alteration of the mouth area.
**Nose and Eyebrow Distortions:** The nose and eyebrows show signs of manipulation, including unnatural shapes, proportions, and structural details. The nostrils are asymmetric or poorly defined, and the shadows cast by the nose are incorrect or missing. The eyebrows are misaligned, asymmetric, or blend unnaturally with the skin and hair, suggesting they were digitally altered or added.
**Hair and Boundary Artifacts:** The hair in the image appears artificial, with an unnatural hairline, unrealistic strand flow, and poor blending with the background and skin. The boundaries of the face, including the jawline and transitions to the neck and hair, are blurry, jagged, or show visible blending artifacts and seams, indicating digital manipulation at the edges.
**Facial Structure and Proportional Issues:** The overall facial structure and proportions appear unnatural, with excessive or unnatural symmetry/asymmetry, disproportionate features, and misaligned landmarks. The face lacks realistic depth and 3D appearance, and the jawline is blurry or unnaturally curved. The facial expression also appears unrealistic or frozen, suggesting a lack of natural movement or structure.
**Lighting, Color, and General Inconsistencies:** The image exhibits inconsistencies in lighting, color, and general visual properties. The lighting direction and quality are inconsistent across the face and with the environment, and shadows and highlights are incorrectly placed or missing. The background shows distortion, and there are inconsistencies in resolution, noise patterns, and color balance between the face and the rest of the image. The presence of general artifacts and unnatural makeup patterns further supports the conclusion of digital manipulation.
**Answer:** fake

**PRPO (Ours):**

**Skin Texture and Detail Abnormalities:** The skin in the image exhibits a lack of natural texture and detail, appearing overly smooth and plastic-like in some areas while lacking realistic pores or exhibiting inconsistent pore distribution. This suggests a synthetic generation or heavy manipulation of the facial surface.
**Eye Abnormalities:** The absence or unnatural appearance of specular highlights (catchlights) in the eyes is a significant indicator of manipulation, as these are crucial for conveying realism and the direction of light in a natural image.
**Teeth Abnormalities:** The teeth show unnatural uniformity in terms of shape, size, color, alignment, and brightness, which is highly uncommon in natural dentition and suggests artificial generation or enhancement.
**Mouth and Teeth Transition Abnormalities:** The unnatural blending of teeth into the lips or gums, along with an irregular gum line or spacing, indicates a failure to accurately render the complex and natural transitions between these facial features.
**Teeth Artifacts:** The presence of pixelation, stretching, smudging, or other artifacts specifically on the teeth points to digital manipulation or generation errors in this region.
**Facial Boundary and Blending Issues:** The boundaries of the face, particularly where it meets the neck, hair, or background, are either unnaturally blurry, poorly defined, or excessively sharp, suggesting an artificial composite or rendering process.
**Overall Image Inconsistencies:** The inconsistency in resolution, pixelation, or sharpness between the face and its surroundings indicates that the face may have been inserted or generated separately from the background, leading to a mismatch in image quality.
**Answer:** fake

*Figure 15.* Qualitative comparison of model reasoning on Example 2.

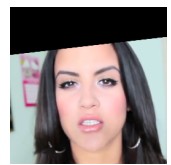

**Ground Truth:** Real

**Question:**

*Analyze this image for any signs of digital manipulation or artificial generation and determine if it is real or fake. Provide your reasoning first, then conclude with a final answer.*

**Qwen2.5-VL-32B:**

The image appears to have shiny, idealized skin which is common in digitally manipulated images or those that have undergone heavy editing, such as smoothing or softening effects. The facial features, particularly the teeth and eyes, appear unnaturally perfect and uniform, which could indicate artificial enhancement. The lighting seems to be overly uniform and lacking in natural imperfections, suggesting digital adjustment for aesthetic purposes. There are no visible signs of natural background texture or context that would provide a sense of authenticity, which could be indicative of a staged or artificial scene. Although not definitive, these characteristics lean towards digital enhancement typical in heavily edited or AI-generated images, although there is still a possibility of this being a very high-quality photo.
**Answer:** fake

**Gemma-3-27B:**

The image exhibits several artifacts strongly suggestive of AI-generation, specifically those commonly seen with faces created by Generative Adversarial Networks (GANs). The skin texture appears overly smooth and lacks realistic microdetails like pores and blemishes. The ears look slightly asymmetric and poorly integrated with the head, which is a common failure point and tell-tale sign of AI-generated imagery. There is a slight blurring around the edges of the face contributing to an artificial look. The lighting is overly uniform and lacks subtle shadows and highlights expected in a real photograph. The mouth's shape and the way it interacts with the face seem slightly off.
**Answer:** fake

**Gemini-2.5:**

The image appears to be a cropped screenshot or frame from a video. The jagged black area at the top left suggests an irregular crop, which is common in user-generated content or screen captures. The overall photographic quality, lighting, and natural appearance of the subject's features do not show typical artifacts associated with advanced AI generation (e.g., distorted features, strange textures, inconsistent lighting, or repetitive patterns). There are no obvious signs of cloning, airbrushing, or other digital manipulations that alter the content of the woman's face or background in a way that suggests fraudulence.
**Answer:** real

**DX-LLaVA (Ours):**

**Eye Characteristics:** The features related to the eyes, such as pupil shape, iris detail, sclera appearance, eyelid shape, gaze direction, eyelashes, and overall eye structure, exhibit typical variations and details expected in a real human face, lacking the common inconsistencies or artificiality often seen in manipulated images.
**Skin and Surface Details:** The skin texture, tone, color, and surface details like blemishes, wrinkles, and highlights appear natural and consistent across different facial regions, without the overly smooth, plastic-like, or patchy appearance characteristic of many deepfakes.
**Mouth and Nose Features:** The features of the mouth (lips, corners, philtrum) and nose (shape, nostrils, shadows, bridge) show natural contours, proportions, and details, avoiding the unnatural shapes, sharp edges, or missing shadows often present in manipulated images.
**Hair and Eyebrows:** The eyebrows and hair, including the hairline, strand flow, texture, and blending with the skin and background, appear realistic and naturally integrated, lacking the artificial sharpness, unnatural blending, or repetitive patterns found in synthetic imagery.
**Facial Structure and Proportions:** The overall facial structure, proportions, symmetry, and the definition of underlying bone and muscle appear natural and consistent with a real human face, without the disproportionate features, misaligned landmarks, or unnatural flatness sometimes observed in deepfakes.
**Lighting, Shadows, and Boundaries:** The lighting, shadows, highlights, and the definition of facial boundaries (jawline, neck, hair) are consistent with the apparent environment and natural physics, lacking the contradictory shadows, unnatural highlights, or blurry/jagged edges that can indicate manipulation.
**Overall Image Consistency and Artifacts:** The image as a whole exhibits consistency in resolution, noise, color, and depth of field. There is an absence of general artifacts, blending issues, or inconsistencies in accessories, clothing, or the background near the face, which are common indicators of digital manipulation.
**Answer:** real

**PRPO (Ours):**

**Eye Characteristics:** The typical appearance of the eyes, including natural catchlights, sclera color and texture, aligned gaze, and normal eye structure, are observed in the image. This consistency with natural human eyes suggests the image is real.
**Skin Texture and Features:** The skin in the image exhibits natural variations in texture and features like blemishes, moles, wrinkles, and creases. The absence of unnatural smoothness, missing features, or inconsistent wrinkles is consistent with a real image.
**Mouth and Teeth Characteristics:** The mouth and teeth display natural characteristics, such as typical number and shape of teeth, realistic lip contour and symmetry, and a normal philtrum shape. The lack of artificial uniformity or distortion in these features supports the image's authenticity.
**Nose and Ear Characteristics:** The nose and ears show natural shapes, proportions, and structural details. The nostrils are well-defined, and the ears are consistent in size and position relative to the face. The absence of unnatural asymmetry or blending issues is typical of a real image.
**Eyebrow Characteristics:** The eyebrows appear naturally aligned and complete, without unnatural blending into the skin or hair. This natural appearance is consistent with an unaltered image.
**Hair Characteristics:** The hair strands exhibit realistic flow, shape, texture, and detail. The hair appears to blend naturally with the background and skin, lacking the artificial or unnatural blending often seen in manipulated images.
**Overall Facial Structure and Symmetry:** The facial structure and symmetry appear within the range of natural human variation. The jawline edges are defined without unnatural blurriness or jaggedness, and the facial expression appears natural and consistent with the context. The absence of excessive or unnatural symmetry or asymmetry supports the image's authenticity.
**Answer:** real

*Figure 16.* Qualitative comparison of model reasoning on Example 3.

