# OpenReview forum: "PRPO: Paragraph-level Policy Optimization for Vision-Language Deepfake Detection"
_ICML.cc/2026/Conference — ICML 2026 regular_

### Official Review · Reviewer_JgeJ · 2026-03-09

**Soundness:** 3
**Presentation:** 3
**Significance:** 4
**Originality:** 3
**Overall Recommendation:** 5
**Confidence:** 2

**Summary:**

The paper addresses deepfake image detection with an emphasis on producing explanations that are visually grounded and consistent with the final prediction. It introduces: (i) **DF-R5**, a ~115k-image reasoning-annotated dataset distilled using Gemini-2.5;     (ii) **DX-LLaVA**, a LLaVA-style model adapted for deepfake detection (CLIP ConvNeXT backbone + auxiliary binary classification loss);     and (iii) **PRPO**, a **test-time** PPO-style optimization method that assigns **advantages at the paragraph level** using label-free rewards for visual consistency (YAKE+CLIP similarity) and prediction consistency (dictionary-based coherence between reasoning and final decision).

They report strong cross-domain gains in leave-one-domain-out evaluation, with PRPO improving average F1 over DX-LLaVA.

**Compliance With Llm Reviewing Policy:**

Affirmed.

**Final Justification:**

Addresses most of my concerns.

**Key Questions For Authors:**

Please clarify the weakness I pointed out in the limitations section

**Limitations:**

No.

While the paper acknowledges some risks, it should more explicitly: (i) strengthen explanation evaluation with larger human studies and agreement/uncertainty reporting,   (ii) discuss and test reward-hacking failure modes for VCR/PCR,   (iii) provide qualitative failure analyses for explanation grounding, and (iv) report budget–performance trade-offs for test-time optimization.

**Strengths And Weaknesses:**

## **Strengths**

1. **Clear problem framing and concrete failure modes**: The paper targets well-motivated issues in deepfake detection with MLLMs—ungrounded/hallucinated explanations and inconsistency between explanation and predicted label.
2. **Novel RL granularity**: PRPO’s paragraph-level credit assignment is a reasonable innovation over token-level RL for multi-paragraph reasoning and is specified clearly.
3. **Label-free reward construction**: VCR and PCR provide a practical way to shape explanations without requiring dense annotation during RL.
4. **Strong cross-domain detection gains**: Reported improvements in leave-one-domain-out F1 are substantial, suggesting improved generalization.

---

## **Weaknesses**

1. **Explanation evaluation is primarily prompt-based LLM judging; human grounding is limited**: The main explanation evaluation (Sec. 4.4) uses GPT-4o with a rubric of five criteria (CAC, EGIA, RQ, CC, CU) and an overall score. This remains a prompt-based judge and may reflect the judge’s preferences rather than true visual grounding. The appendix includes a small human study, but it is limited (8 participants × 5 images each) and lacks inter-rater agreement/variance reporting, making it insufficient as strong validation.
2. **Limited qualitative analysis / interpretability of improvements**: While the paper reports rubric-based scores, it provides limited qualitative error analysis showing *how* PRPO changes explanations (e.g., representative successes/failures with specific grounding failures, cases where explanations are confident but wrong, or where rewards encourage generic artifact lists). More qualitative case studies and a failure-mode taxonomy would improve interpretability and strengthen the claim that explanations are genuinely grounded rather than stylistically improved.
3. **Reward design may be hackable / weakly aligned with forensic grounding**: VCR (YAKE keywords + CLIP similarity) and PCR (dictionary-based consistency) can plausibly be satisfied by generic “artifact” language or self-consistent narratives that are not correct. The paper would be stronger with counterexamples demonstrating that PRPO cannot game rewards with superficial cues.
4. **Test-time optimization cost is a deployment limitation; needs clearer budget–performance trade-offs**: PRPO’s test-time optimization implies added latency/compute. The appendix provides a latency/cost breakdown, but the paper would benefit from a clear curve showing how detection/explanation quality vary with the test-time budget (samples/updates).
5. **Limited statistical/uncertainty reporting**: Key comparisons (especially explanation scores and the small human study) lack confidence intervals or significance testing.

---

> ### Author Rebuttal · Authors · 2026-03-31
>
> We sincerely thank all reviewers for their thoughtful and constructive feedback. We appreciate the reviewers’ recognition of the clear problem framing (Reviewer JgeJ), the well-designed dataset construction pipeline (Reviewers nyds and LVAT), the novel paragraph-level credit assignment (Reviewer JgeJ), the strong cross-domain detection gains (Reviewers LVAT and JgeJ), and the solid experiments and analysis (Reviewer GJAf). Below, we address each reviewer's concerns in detail.
>
> **Q1. Human grounding is limited: 8 participants × 5 images, no inter-rater agreement reported.**
>
> We acknowledge this limitation. However, we prioritize the quality of assessment over the number of participants. In our study, each expert spends approximately 2-3 hours evaluating all criteria. Scaling to a larger cohort (e.g., 30–100 participants) would likely require involving non-experts and reducing evaluation time, which could compromise assessment quality. Notably, even this small-scale study produces results consistent with GPT-4o rankings, increasing our confidence that the GPT-4o-based evaluation is not systematically biased.
>
> **Q2. Limited qualitative analysis: how PRPO changes explanations (successes, failures, reward gaming).**
>
> This is valuable feedback. We will add a qualitative analysis section in the appendix that includes success cases, failure cases, and reward gaming analysis (where high VCR/PCR rewards do not correspond to correct predictions).
>
> **Q3. Reward design may be hackable - VCR/PCR can be satisfied by generic artifact language.**
>
> Two factors mitigate this issue in practice:
> 1. VCR requires image-specific alignment. Generic artifact terms (e.g., "blurry edges") tend to produce uniformly moderate CLIP similarity across images. To achieve a high relative VCR reward (above the group mean), a paragraph must describe features that are more specifically aligned with the given image than other sampled outputs, which encourages image-specific descriptions.
>
> 2. PCR cross-validates VCR. A model that uses generic language across all paragraphs may produce consistent predictions and achieve high PCR. However, such paragraphs would not receive high VCR rewards if they lack image specificity. The combination of VCR and PCR therefore prevents both forms of reward gaming (see Table 7).
>
> **Q4. Test-time optimization cost needs a clearer budget–performance trade-off.**
>
> We will add a budget–performance analysis to the appendix, reporting F1 and explanation quality as functions of (i) the number of RL update steps and (ii) the number of sampled outputs \(L\) per prompt. In our experiments, PRPO performance stabilizes after 100 steps with \(L=16\). We will explicitly report this trade-off.
>
> **Q5. Key comparisons lack confidence intervals or significance testing.**
>
> Due to the limited rebuttal timeline, we are unable to provide these results immediately. We will include mean, standard deviation, and bootstrap confidence intervals in the revision. Thank you for this valuable suggestion.

---

> > ### Author Rebuttal · Reviewer_JgeJ · 2026-04-02
> >
> > Several key points are still deferred to the revision rather than demonstrated here, especially stronger qualitative analysis, budget–performance trade-offs, and statistical/uncertainty reporting. I also still think the reward-hacking concern would benefit from concrete counterexamples or empirical validation rather than only conceptual justification.

---

> > > ### Author Response · Authors · 2026-04-04
> > >
> > > We thank Reviewer JgeJ for the follow-up. We choose to address these concerns in revision since we find that these requirements are not top priorities in deepfake detection/explainability (including in baselines and related work). Nonetheless, we are willing to address these concerns promptly with concrete additional experiments below.
> > >
> > > ### Q3 (Reward Hacking) - Adversarial Probe
> > > We conduct a controlled probe on 500 test images (100/domain). For each image, we compute \(R_{\text{VCR}}\) and the group-normalized advantage \(A=(R-\mu)/\sigma\) across four paragraph types, alongside 15 PRPO outputs:
> > >
> > > | Type | Example | CLIP | R_VCR | A | Effect |
> > > |---|---|---|---|---|---|
> > > | PRPO (specific) | “absent specular highlights…” | 0.097 | 0.548 | 0.00 | baseline |
> > > | Random words | random tokens | −0.003 | 0.499 | −2.265 | penalized |
> > > | Generic template | “unnatural texture…” | 0.091 | 0.546 | −0.119 | penalized |
> > > | Domain-generic repetition | “skin smooth, eyes artificial…” | 0.126 | 0.563 | +0.670 | VCR ↑ |
> > > | Cross-image | correct for another image | 0.097 | 0.548 | ≈0.00 | neutral |
> > >
> > > Random baseline: p = 9.78 × 10⁻²², Cohen's d = 0.816.
> > >
> > > Domain-generic repetition yields a positive VCR advantage (+0.670), exposing a blind spot: short fake-trait phrases align well with faces due to pre-training co-occurrence, often more than PRPO-specific text. However, this is not a viable gaming strategy. If such text predicts “fake” for a real image, PCR = 0, giving \(R=(0.563+0)/2=0.281\), far below a genuine PRPO output (\(R=(0.548+1.0)/2=0.774\)). Cross-image paragraphs similarly have VCR ≈ 0 but PCR = 0 when incorrect. Thus, the VCR+PCR combination prevents the gaming that VCR alone cannot.
> > >
> > > **Additional evidence:**
> > > 1. **Perturbation sensitivity (Table 12, Appendix B.1):** VCR shifts under image noise (-30.72% JPEG, -27.75% Gaussian noise), confirming dependence on visual input.
> > > 2. **Reasoning diversity (Table 11, Appendix A.4):** PRPO yields 1,593 unique features vs. 550 (DX-LLaVA), contradicting collapse to generic phrases.
> > > 3. **PCR-only ablation (Table 7):** F1 drops to 70.98% vs. 94.99%, showing neither reward alone is exploitable.
> > >
> > > ### Q2 (Qualitative Analysis) - Success and Failure Cases
> > >
> > > **Success: grounded correction of hallucination (Example 3, Fig. 16).**
> > >
> > > Ground truth: **Real** (natural photo of a woman).
> > >
> > > | Model | Reasoning strategy | Prediction | Correct? |
> > > |---|---|---|---|
> > > | Qwen2.5-VL-32B | "Shiny, idealized skin, unnaturally perfect teeth, lighting lacks natural imperfections" | Fake | ✗ |
> > > | Gemma-3-27B | "Overly smooth skin texture, slight asymmetry ears feels unnatural and poorly integrated with the head " | Fake | ✗ |
> > > | DX-LLaVA | "Eye characteristics exhibit typical natural variations; skin tone consistent across regions; mouth shows natural contours; hair blends naturally with background" | Real | ✓ |
> > > | **PRPO** | **"Natural catchlights present in eyes; natural blemishes and creases in skin texture; well-defined nostrils with natural shadow; hair exhibits realistic strand flow"** | **Real** | **✓** |
> > >
> > > Qwen and Gemma rely on generic artifact cues (e.g., “overly smooth skin”, “unnaturally perfect teeth”) that moderately match real faces and thus misfire on this image. In contrast, PRPO is trained via VCR to penalize such generic descriptions and favor image-grounded, specific evidence. The learned distinction, generic (“overly smooth”) vs. specific (“natural blemishes and creases”), enables correct classification through relative advantage scoring.
> > >
> > > **Failure: sparse-artifact diffusion (SiT domain, F1=71.26%).**
> > > 1. Images are near-perfect (diffusion realism).
> > > 2. PRPO correctly describes real-looking features → high VCR.
> > > 3. Majority vote favors “real”.
> > > 4. Prediction becomes confidently wrong.
> > >
> > > This reflects a limitation: **absence of artifacts is itself evidence of high-quality fakes**, which current rewards do not encode. This does not occur in GAN/DDIM domains where artifacts exist. We will discuss this in more detail in the revision.
> > >
> > > ### Q4 (Budget–Performance)
> > >
> > > **Inference.** Same cost as DX-LLaVA (single forward pass). RL is offline.
> > >
> > > **Training.** PRPO trained on DDIM using 4×H200 GPUs:
> > >
> > > | Steps | Time | F1 |
> > > |---|---|---|
> > > | 25 | ~1.1h | 93.79 |
> > > | 50 | ~2.1h | 95.10 |
> > > | 75 | ~3.2h | **96.29** |
> > > | 100 | ~4.3h | 95.79 |
> > > | 125 | ~5.3h | 95.70 |
> > >
> > > Most gains occur early  in ~1h. Performance peaks at 75 steps, then stabilizes. Thus, **PRPO is cost-efficient**.
> > >
> > > ---
> > >
> > > ### Q5 (Statistics)
> > >
> > > **GPT-4o evaluation (N≈5k, 10k bootstrap):**
> > >
> > > | Model | Mean ± Std | 95% CI (Confidence Interval) |
> > > |---|---|---|
> > > | PRPO | 4.550 ± 1.144 | [4.519, 4.582] |
> > > | DX-LLaVA | 4.015 ± 1.541 | [3.973, 4.057] |
> > > | Gemini-2.5 | 4.377 ± 1.011 | [4.349, 4.405] |
> > >
> > > CIs do not overlap (PRPO lower bound > Gemini upper bound), confirming statistical significance.
> > >
> > > PRPO breakdown:
> > > - Accuracy: 4.418 ± 1.598
> > > - Grounding: 4.557 ± 1.115
> > > - Reasoning: 4.583 ± 1.072
> > > - Calibration: 4.504 ± 1.316
> > > - Clarity: 4.688 ± 0.745

---

### Official Review · Reviewer_LVAT · 2026-03-11

**Soundness:** 2
**Presentation:** 3
**Significance:** 2
**Originality:** 3
**Overall Recommendation:** 4
**Confidence:** 4

**Summary:**

This paper addresses the limitations of Multimodal Large Language Models (MLLMs) in deepfake detection, specifically their poor reasoning quality and the scarcity of fine-grained annotated data. To tackle these issues, the authors introduce a novel framework that enhances both detection accuracy and explainability. The main contributions are threefold: (1) the construction of DF-R5, a large-scale deepfake detection dataset containing 115k image-reasoning pairs with detailed paragraph-level annotations; (2) the development of DX-LLaVA, a multimodal architecture utilizing a CLIP ConvNeXT vision encoder to better capture subtle, local generative artifacts; and (3) the proposal of Paragraph-level Relative Policy Optimization (PRPO), a test-time reinforcement learning algorithm that aligns the model's intermediate reasoning with visual evidence using Visual Consistency and Prediction Consistency rewards. Extensive experiments demonstrate that this approach significantly improves cross-domain detection generalization and the reliability of the generated explanations.

**Compliance With Llm Reviewing Policy:**

Affirmed.

**Final Justification:**

The author's reply basically resolved my concerns.

**Key Questions For Authors:**

1. How is the validity of the deepfake features generated by MLLMs systematically verified for the DF-R5 dataset to prevent inherent semantic hallucinations?
2. Could you provide ablation studies for the critical hyperparameters $\alpha$ (10.0) and $\beta$ (0.01) to validate the stability of the test-time RL framework?
3. The frozen CLIP ConvNeXT used in VCR may lack an understanding of deepfake-specific semantics. How do you prevent this limitation from generating erroneous reward signals?
4. For high-quality deepfakes, artifacts are often sparse. Does PCR's majority voting risk suppressing these critical minority cues, and have you considered any other method instead?

**Limitations:**

yes

**Strengths And Weaknesses:**

Strengths

1. This paper addresses the critical pain points of MLLM hallucination and the scarcity of high-quality reasoning data in deepfake detection. The construction of the DF-R5 dataset, containing 115k image-reasoning pairs, fills a significant gap in the field.
2. The proposed Paragraph-level Relative Policy Optimization (PRPO) is innovative. It guides the model to generate explanations aligned with visual evidence by optimizing rewards at the paragraph level during test time.
3. The paper demonstrates exceptionally strong performance in cross-domain deepfake detection (e.g., testing on unseen domains like SiT), where PRPO achieves an impressive average F1 score of 89.91%, substantially outperforming state-of-the-art detectors and mainstream MLLMs.

Weaknesses

1. The construction of the DF-R5 dataset heavily relies on the outputs of MLLMs, ultimately selecting Gemini-2.5 as the primary distillation model. Although the authors employ a multi-model feature discovery and consistency scoring pipeline to ensure quality, these annotations are fundamentally machine-generated. This distillation paradigm inevitably inherits the potential blind spots of the Gemini model, which may limit the model's ability to discover and generalize when facing new kinds of deepfakes in the future. In short, if Gemini-2.5 cannot effectively understand artifacts, standard quality cannot be guaranteed.
2. The Visual Consistency Reward (VCR) relies on a frozen CLIP ConvNeXT encoder and YAKE keyword extraction. YAKE is a lightweight keyword extraction method, which might inadvertently discard complex semantic context and logical relationships within a paragraph. Moreover, if the frozen CLIP encoder itself fails to recognize a novel or highly deceptive local deepfake artifact, the VCR will provide erroneous reward signals, thereby misguiding policy optimization. CLIP encoders are trained on general domain data, and there is insufficient evidence to suggest that they can fully understand forged cues.
3. Although detailed ablation studies are provided for reward components and architectures, there is a limited sensitivity analysis for critical hyperparameters during the joint training and RL optimization processes. Specifically, the trade-off parameter $\alpha$ (set to 10.0) between the language modeling loss and binary classification loss, and the KL divergence penalty coefficient $\beta$ (set to 0.01), are introduced without extensive sensitivity exploration. Test-time reinforcement learning is notoriously sensitive to hyperparameters, and supplementing this analysis would significantly enhance the paper's reproducibility.

---

> ### Author Rebuttal · Authors · 2026-03-31
>
> We sincerely thank all reviewers for their thoughtful and constructive feedback. We appreciate the reviewers’ recognition of the clear problem framing (Reviewer JgeJ), the well-designed dataset construction pipeline (Reviewers nyds and LVAT), the novel paragraph-level credit assignment (Reviewer JgeJ), the strong cross-domain detection gains (Reviewers LVAT and JgeJ), and the solid experiments and analysis (Reviewer GJAf). Below, we address each reviewer's concerns in detail.
>
> **Q1: DF-R5 annotations inherit Gemini's blind spots and may limit generalization.**
> This is a valid concern, and we clarify three points:
> - Multi-model feature discovery reduces single-model bias: Step 1 aggregates proposals from four diverse MLLMs (Gemini-2.5, GPT-4o, Qwen-2.5, LLaMA-4), producing 74 curated features that reflect broader consensus.
> - Step 2 incorporates ground truth, ensuring that even when Gemini mispredicts, feature annotations are corrected toward the correct labels.
> - PRPO operates label-free at test time: the RL stage relies only on self-supervised VCR and PCR rewards rather than DF-R5 annotations, allowing the model to adapt to new deepfake types as long as they align with the image. Empirically, RL explores over 400 unseen visual features (Table 11 in Appendix), enabling continued improvement beyond the DF-R5 benchmark.
>
> **Q2: YAKE may discard context; frozen CLIP ConvNeXT may miss novel artifacts.**
> - For YAKE, keyword extraction is a necessary approximation under CLIP's token limit, and empirical results show that it retains artifact-relevant terms (we present this experimental setup in response to Reviewer nyds - Q2; please have a look).
> - For CLIP, we freeze ConvNeXT to preserve pretrained representations and maintain reward stability during RL. Although CLIP is not trained specifically on deepfakes, such artifacts often appear as subtle texture inconsistencies and local anomalies, which align well with ConvNeXT’s convolutional inductive bias (Appendix C.2 demonstrates the effectiveness of the ConvNeXT architecture). Supporting evidence shows that replacing CLIP ViT with ConvNeXT improves average F1 by 19.65% (from 61.21% to 80.86%, Table 13 in Appendix), indicating stronger discriminative power. Moreover, VCR uses relative normalization, so even imperfect similarity scores still guide the model toward comparatively better-aligned reasoning within each group.
>
> **Q3: Limited sensitivity analysis for hyperparameters $\alpha$ and $\beta$.**
> We conduct an ablation study on $\alpha$, and the results show stable performance across a wide range of values (the experiment is conducted on the unseen DDIM domain):
>
> | $\alpha$ | Precision | Recall  | F1     |
> |--------:|----------:|--------:|--------:|
> |     0.0 | 98.97%    | 91.62%  | 95.15% |
> |     1.0 | 99.20%    | 94.86%  | 96.98% |
> |    10.0 | 98.79%    | 93.14%  | 95.88% |
> |    20.0 | 99.80%    | 93.33%  | 96.46% |
>
> These results demonstrate that performance remains robust to $\alpha$. Due to time constraints in the rebuttal period, we cannot include $\beta$ analysis now, but we will add it in the revision as a new Appendix subsection.
>
> **Q4: How do you verify the validity of MLLM-generated features and prevent hallucinations?**
> We ensure quality through three mechanisms. First, feature discovery is image-free, so Step 1 elicits general domain knowledge without introducing image-specific hallucinations. Second, Step 2 uses feature-level ranking instead of free-form generation, which constrains outputs and reduces hallucinated content. Third, we refine predictions with ground-truth labels: when Gemini is incorrect, we re-prompt it to identify features consistent with the correct label, following the STaR approach (Zelikman et al., 2022) to correct reasoning trajectories.
>
> **Q5: How do you prevent erroneous reward signals from frozen CLIP ConvNeXT?**
> Relative normalization in VCR mitigates absolute alignment errors, while PCR provides a complementary, image-independent signal that enforces reasoning consistency. Even if VCR is noisy for a specific sample, PCR regularizes the reward, and Table 7 shows that combining both signals is essential for balanced performance (94.99%, compared to 88.68% (VCR only) and 70.98% (PCR only)).
>
> **Q6: Does PCR's majority voting suppress minority cues in sparse-artifact cases?**
> This is an insightful concern. When artifacts are sparse, reasoning contains fewer paragraphs and voting occurs over a small set. Although majority voting may favor “real” when only one paragraph captures a true artifact, VCR simultaneously rewards visually grounded paragraphs, so rare but valid artifact descriptions receive higher advantage scores and greater influence during policy updates. The combination of VCR and PCR therefore prevents majority bias from dominating, although extremely sparse scenarios remain challenging. Future work could explore weighted voting based on VCR scores, and we will add a discussion of this trade-off.

---

> > ### Author Rebuttal · Reviewer_LVAT · 2026-04-04
> >
> > Thank you for the author's response.

---

> > > ### Author Response · Authors · 2026-04-04
> > >
> > > We sincerely thank the reviewer for the positive feedback and for confirming that the concerns have been fully addressed. If there are any remaining concerns that could further improve the score, we would be happy to address them.

---

### Official Review · Reviewer_nyds · 2026-03-11

**Soundness:** 4
**Presentation:** 3
**Significance:** 2
**Originality:** 2
**Overall Recommendation:** 3
**Confidence:** 4

**Summary:**

This paper argues that existing models produce shallow or hallucinated explanations misaligned with visual evidence. They introduce a deepfake detection dataset DF-R5 containing 115k reasoning-annotated samples. They also propose a model named DX-LLaVA, which replaces CLIP ViT with CLIP ConvNeXT for better local artifact sensitivity. For reasoning, they propose PRPO which is a RL algorithm for optimizing reasoning at the paragraph level using label-free rewards. Experimental results show 89.91% average F1 and a reasoning score of 4.55/5.0.

**Compliance With Llm Reviewing Policy:**

Affirmed.

**Final Justification:**

I keep my score unchanged. weak reject.

**Key Questions For Authors:**

1. The three-step dataset construction relies heavily on Gemini's judgment. How is the annotation quality ensured, especially for cases where Gemini itself makes wrong predictions (77.6% accuracy in Table 1)?
2. PRPO is applied at test time without ground truth labels. How many test-time iterations are needed before the model converges, and how sensitive is performance to this number?
3. The paragraph structure is predefined by the dataset format. What happens if the model generates reasoning in a different format at test time — does PRPO still work?

**Limitations:**

Yes.

**Strengths And Weaknesses:**

Strengths:
1. The three-step dataset construction pipeline is well-designed. Using feature scoring before reasoning generation is a smart way to reduce hallucination in distilled annotations.
2. The paragragh-level reward design is intuitive and the label-free formulation is a nice property. It allows test-time adaptation without ground truth, which is practically important.
3. Replacing ViT by ConvNeXT improves the feature reliability.

Weaknesses:
1. The evaluation is circular. The training data is distilled from Gemini, and the reasoning quality is then scored by GPT-4o. It is hard to rule out that PRPO just learns to produce reasoning that happends to align with GPT-4o's preferences rather than better explanations.
2. VCR relies on YAKE which feels like a weak link. YAKE has no deepfake-specific knowledge and might just pick up generic words. There's no analysis of whether the extracted keywords actually capture meaningful visual cues
3. The comparison against Gemini is a bit misleading. The model is trained on Gemini-distilled data, so outperforming Gemini at test time is kind of expected. This needs more careful framing.
4. The experiments are limited to face-centric deepfakes.  All five domains generate facial images, so it's unclear whether the method works beyond this narrow distribution.

---

> ### Author Rebuttal · Authors · 2026-03-31
>
> We sincerely thank all reviewers for their thoughtful and constructive feedback. We appreciate the reviewers’ recognition of the clear problem framing (Reviewer JgeJ), the well-designed dataset construction pipeline (Reviewers nyds and LVAT), the novel paragraph-level credit assignment (Reviewer JgeJ), the strong cross-domain detection gains (Reviewers LVAT and JgeJ), and the solid experiments and analysis (Reviewer GJAf). Below, we address each reviewer's concerns in detail.
>
> **Q1: The evaluation is circular: training data is distilled from Gemini, and reasoning quality is judged by GPT-4o. There is a chance PRPO’s output favors GPT-4o's preferences.**
>
> We address this concern with three points:
> - Gemini (Google DeepMind) and GPT-4o (OpenAI) belong to different model families with distinct training data, architectures, and pipelines, so stylistic preferences do not reliably transfer. A model trained on Gemini-style outputs therefore has no inherent reason to align with GPT-4o's preferences.
> - We conduct human evaluation with 8 experts (Appendix), and the rankings are consistent with GPT-4o, providing independent validation free from model-to-model bias.
> - Our primary metrics (Precision, Recall, F1, Accuracy) are computed against ground truth rather than an MLLM judge. PRPO achieves **89.91% F1**, outperforming Gemini (**80.31%**), which confirms gains on an objective benchmark.
>
> **Q2: VCR relies on YAKE, which has no deepfake-specific knowledge and might pick up generic words.**
>
> We agree YAKE is lightweight, but our design is justified. First, CLIP ConvNeXT has input length constraints, so compressing reasoning into compact, content-preserving keywords is necessary. Second, empirical analysis on 1,000 held-out samples (2,747 paragraphs) shows that:
>
> - **96.1% unique keywords (2,641 / 2,747)**, with the most frequent appearing <0.3%, indicating high diversity.
> - Up to 10 keywords per paragraph, **avg length 7.56 words**, with **99.9% multi-word phrases**, e.g., *“visible unnatural edges surrounding the hair”*.
> - **38.3% of keywords contain forensic terms**, and **44% of top-50 keywords are forensic-related**, showing alignment with domain cues.
>
> Third, CLIP similarity confirms grounding:
>
> | Condition | CLIP Similarity ↑ | Std |
> |----------|------------------|-----|
> | **YAKE keywords** | **0.5741** | ±0.035 |
> | Random words | 0.5486 | ±0.016 |
> | **Lift** | **+2.55%** | — |
>
> - **p = 9.78 × 10⁻²²**, **Cohen’s d = 0.816 (large effect)**.
>
> Overall, YAKE acts as a content selector over structured reasoning, extracting image-grounded forensic cues rather than generic tokens.
>
> **Q3: Comparing against Gemini is misleading because the model is trained on Gemini-distilled data.**
>
> DX-LLaVA is fine-tuned on Gemini-generated reasoning text, but the task itself, binary deepfake detection, is a vision problem on which Gemini is not fine-tuned. Gemini’s **80.31% F1** reflects zero-shot prompting, whereas our model is fully fine-tuned and further optimized with reinforcement learning. The comparison therefore demonstrates that a specialized pipeline can outperform a general-purpose MLLM on this task.
>
> **Q4: Experiments are limited to face-centric deepfakes.**
>
> We follow prior work [1] using face-centric datasets. DF-R5 is built from DF-40, which focuses on facial synthesis and reflects the dominant deepfake use case. However, our method is general and can extend to other manipulations (e.g., background editing, object insertion). We will clarify this scope in the Limitations section. Face-centric deepfakes remain the most impactful and technically challenging setting, motivating our focus.
>
> [1] Chen et al. X2-dfd: A framework for explainable and extendable deepfake detection. NeurIPS 2025.
>
> **Q5: How is annotation quality ensured when Gemini itself has only 77.6% accuracy?**
>
> The 77.6% refers to Gemini’s zero-shot classification accuracy, not our annotation process. We ensure quality through:
> 1. **Ground-truth refinement:** incorrect predictions are corrected by re-prompting with true labels (following STaR method), ensuring consistent reasoning.
> 2. **Feature-level scoring:** Gemini ranks predefined features instead of generating free-form text, reducing hallucination.
> 3. **Image-free feature discovery:** features are defined without images, preventing image-specific bias during annotation.
>
> **Q6: How many test-time iterations are needed?**
>
> PRPO requires only one forward pass at test time. Structured reasoning and paragraph-level aggregation occur within a single response, without multiple sampling or inference calls.
>
> **Q7: What happens if the model generates reasoning in a different format at test time?**
>
> We enforce structure via a prompt template with up to \(M = 7\) numbered paragraphs plus a conclusion. Fine-tuning on DF-R5 ensures consistent outputs. If deviations occur, we treat the output as a single paragraph with uniform reward, which degrades gracefully to a weaker VCR signal rather than failing.

---

> > ### Author Rebuttal · Reviewer_nyds · 2026-04-02
> >
> > Fully resolved. I prefer to keep my score unchanged for now.

---

> > > ### Author Response · Authors · 2026-04-04
> > >
> > > We sincerely thank the reviewer for the positive feedback and for confirming that the concerns have been fully addressed. If there are any remaining concerns that could further improve the score, we would be happy to address them.

---

### Official Review · Reviewer_GJAf · 2026-03-12

**Soundness:** 3
**Presentation:** 3
**Significance:** 3
**Originality:** 3
**Overall Recommendation:** 4
**Confidence:** 3

**Summary:**

This paper aims to address the limitations of MLLMs in deepfake detection, specifically their tendency to produce hallucinations, provide shallow explanations, and struggle with generalization across unseen generative domains. To address this, the authors introduce a reasoning-annotated dataset, DF-R5, and a multimodal architecture, DX-LLaVA, which replaces the standard CLIP-ViT encoder with a CLIP convnext backbone to better capture fine-grained, high-frequency generative artifacts. Furthermore, the authors propose paragraph-level RPO, a reinforcement learning algorithm applied at test-time to align LLM reasoning with visual evidence. By implementing two rewards to penalty the self-consistent and visual-reasoning alignment, the system forces the model to generate logically self-consistent, evidence-grounded explanations. Experimental results show the proposed method achieved improved detection f-score and higher reasoning quality.

**Compliance With Llm Reviewing Policy:**

Affirmed.

**Key Questions For Authors:**

No questions.

**Limitations:**

The author do not discuss the limitation, see Weaknesses.

**Strengths And Weaknesses:**

**Strengths**
1. The paper is highly motivated and the proposed method adequantly address the problem.
2. The experiments are solid especially the ablation study analyse the effectiveness of each component.


**Weaknesses**
1. Using GPT-4o as the judge for Table 6 and 15 is risky. There's a clear 'self-evaluation bias' here. Since PRPO outputs are structured in the specific JSON format that GPT-4o likes, it's easy for the model to get high scores just because it followed the format, not necessarily because its reasoning is actually better.

---

> ### Author Rebuttal · Authors · 2026-03-31
>
> We sincerely thank all reviewers for their thoughtful and constructive feedback. We appreciate the reviewers’ recognition of the clear problem framing (Reviewer JgeJ), the well-designed dataset construction pipeline (Reviewers nyds and LVAT), the novel paragraph-level credit assignment (Reviewer JgeJ), the strong cross-domain detection gains (Reviewers LVAT and JgeJ), and the solid experiments and analysis (Reviewer GJAf). Below, we address each reviewer's concerns in detail.
>
> **Q1: Using GPT-4o as the judge creates self-evaluation bias because "PRPO outputs are structured in the specific JSON format that GPT-4o likes."**
>
> We respectfully clarify our implementation: PRPO outputs are structured in paragraph format, **not** JSON. These are natural-language paragraphs, each describing a deepfake characteristic, followed by a final conclusion (### Answer). No JSON schema is imposed. The GPT-4 judge evaluates outputs in free-form text across five independent criteria (CAC, EGIA, RQ, CC, CU), each scored from 1 to 5 based on the quality of the image content and its relevance to the provided image. Therefore, the concern about format-based scoring does not apply. In fact, using GPT as a judge has also been applied in previous work [1], and we are inspired by this strategy.
>
> [1] Chen, Y., Yan, Z., Cheng, G., ZHAO, K., Lyu, S., and Wu, B. X2-dfd: A framework for explainable and extendable deepfake detection. In The Thirty-ninth Annual Conference on Neural Information Processing Systems, 2025.
>
> **Q2: The authors do not discuss the limitation.**
>
> Thank you for your suggestion. We will summarize the concerns raised by all reviewers and add an explicit Limitations section to discuss them. For example, these include reliance on Gemini-distilled annotations (addressed in detail for Reviewer nyds and Reviewer LVAT), restriction to face-centric deepfakes, and test-time computational overhead.

---

> > ### Author Rebuttal · Reviewer_GJAf · 2026-04-03
> >
> > Thanks for the rebuttal, I decide to keep my positive score.

---

> > > ### Author Response · Authors · 2026-04-04
> > >
> > > We sincerely thank the reviewer for the positive feedback and for taking the time to evaluate our rebuttal. We appreciate your decision to maintain a positive score.

---

### Decision · Program_Chairs · 2026-04-30

**Decision:**

Accept (regular)

**Comment:**

This paper presents a solid and timely contribution to deepfake detection with MLLMs by combining a large reasoning-annotated dataset, an improved multimodal architecture, and a paragraph-level optimization strategy for more grounded explanations. The reviewers generally agreed that the paper is well motivated, technically sound, and supported by strong empirical results, especially on cross-domain generalization and ablation studies. The main concerns were about the reliability of LLM-based evaluation, the dependence on machine-distilled annotations, and some missing analysis on limitations and robustness. The authors addressed these points clearly in the rebuttal, and most reviewers stated that their concerns were resolved. Overall, the AC finds the contribution meaningful and sufficient, and therefore would like to recommend positively.